# Climate change threatens Chinook salmon throughout their life cycle

Lisa G. Crozier [1]✉, Brian J. Burke [1], Brandon E. Chasco [1], Daniel L. Widener [2] & Richard W. Zabel [1]

Widespread declines in Atlantic and Pacific salmon (*Salmo salar* and *Oncorhynchus* spp.) have tracked recent climate changes, but managers still lack quantitative projections of the viability of any individual population in response to future climate change. To address this gap, we assembled a vast database of survival and other data for eight wild populations of threatened Chinook salmon (*O. tshawytscha*). For each population, we evaluated climate impacts at all life stages and modeled future trajectories forced by global climate model projections. Populations rapidly declined in response to increasing sea surface temperatures and other factors across diverse model assumptions and climate scenarios. Strong density dependence limited the number of salmon that survived early life stages, suggesting a potentially efficacious target for conservation effort. Other solutions require a better understanding of the factors that limit survival at sea. We conclude that dramatic increases in smolt survival are needed to overcome the negative impacts of climate change for this threatened species.

[1] Fish Ecology Division, Northwest Fisheries Science Center National Marine Fisheries Service, National Oceanic and Atmospheric Administration, Seattle, WA, USA. [2] Ocean Associates, Inc. Under contract to Northwest Fisheries Science Center National Marine Fisheries Service, National Oceanic and Atmospheric Administration, Seattle, WA, USA. ✉email: Lisa.Crozier@noaa.gov

Widespread declines of wild salmon[1-5] have negatively affected fisheries, cultural heritage for indigenous tribes[6], and other marine species, including endangered Southern Resident Killer Whales[7]. Currently, the majority of Atlantic and Pacific salmon and steelhead (*Salmo salar* and *Oncorhynchus* spp.) in the conterminous U.S. are threatened with extinction[8]. Over the past century, declines were driven by overfishing, migration barriers, water diversions, habitat loss, salmon farms, and hatcheries, as well as regime shifts in marine ecosystems[9]. Retrospective analyses also show strong relationships between climate indices and salmon performance[10,11]. Looking toward the future, indirect and qualitative assessments point to anthropogenic climate change as an additional overriding threat for salmon in the North Atlantic and California Current[12-15]. How to mitigate this threat is therefore a primary concern among conservation organizations and management agencies.

Previous population models that have used global climate model (GCM) projections have focused on drivers in freshwater life stages only (e.g., stream temperature, winter flooding, and drought)[16-18]. While these are useful for evaluating restoration actions within those contexts, they completely ignore the large impacts of climate change on the marine stage. On the other hand, GCM projections related to marine survival have been used primarily to inform niche-based models that forecast future habitat for salmon generally rather than for specific populations[19,20]. Although marine climate indices such as the Pacific Decadal Oscillation (PDO)[21] and Atlantic Multidecadal Oscillation (AMO)[4] have been tightly linked to survival, they cannot readily be used in projections of future climate because these phenomena are not consistently reproduced in GCM projections for complex reasons[22,23]. To produce a novel, robust analysis of climate change impacts, our analysis focused on climate drivers with more reliable performance in GCM projections that were also closely correlated with fish survival.

A more general limitation of previous models has been a failure to account for large-scale climate forcing that affects multiple life stages and food webs simultaneously, potentially compounding climate effects across developmental stages. Accounting for the correlation structure of climate effects over the full life cycle is especially important for migratory species with complex life histories[15,24]. We acknowledge that relationships between survival and climate are often nonstationary[25], so we made the correlation structure of environmental drivers explicit and flexible enough to incorporate future change. This is a new approach to downscaling climate projections in multiple environments.

These limitations contribute to three main reasons why existing approaches for modeling biological impacts of climate change are inadequate for informing or evaluating potential management actions for salmon conservation. Similar limitations apply to other species that are migratory or have complex life histories.

First, proposed management actions are usually focused on conditions in freshwater, but accounting for "carryover" effects from freshwater to marine life stages is essential for the evaluation of such actions. Carryover effects occur when the previous history of an individual affects its performance in a subsequent life stage[26]. For example, the timing of migration from freshwater to the ocean and back to freshwater are key determinants of salmon survival in other life stages, and one of the most climate-sensitive traits[27-30]. Migration timing is also a key element in multiple management actions, especially those involving the hydrosystem[31] and fisheries[32]. Therefore, quantification of carryover effects that will be affected by climate change is essential for evaluating the net benefit of proposed actions to protect endangered species.

Second, present models of survival in the salmon marine stage rely on climate indices, which cannot be linked directly to global climate model (GCM) projections. Therefore, it is impossible to conduct formal analyses of how alternative carbon emission scenarios or other anthropogenic actions to mitigate climate change might affect the timing of declines in marine survival. Nor is it possible to quantify uncertainty in modeled projections across GCMs and thus take full advantage of the *Coupled Model Intercomparison Project*, which represents the major advances of global climate modeling in recent decades[33,34].

Third, approaches that are currently available to account for climate impacts on freshwater and marine life stages use independent downscaling methods for the two environments. Terrestrial downscaling methods usually employ statistical or dynamical downscaling of temperature and precipitation, which are then fed into hydrological models. Statistical downscaling is an efficient way to explore many alternative climate projections and characterize model uncertainty at many steps in the modeling chain[35]. In contrast, a common approach to marine downscaling is to integrate GCM output into regional ocean models (ROMS), which in practice are only available for very few GCM projections[34]. As a result, these methods often rely on projections from different GCMs and are not consistent in characterizing potential model biases, and thus uncertainty in climate projections. Moreover, these multiple GCMs are not temporally linked, which impedes the ability to account for carryover effects from one life stage to the next[36].

We attempted to address each of these difficulties by developing a modeling approach with flexible and explicit mechanisms to account for the correlation structure among all climate drivers. We also used a multimodel approach to indirectly account for the change in the relationship between climate drivers and ecological responses. Finally, we allowed the initiation timing of juvenile and adult migration to vary with environmental conditions, which subsequently affected both smolt and adult migration survival. Timing also affected the probability that juvenile fish would be transported; transport by barge through the hydrosystem is a mitigation action that has fixed start and stop dates each year. Three factors—migration timing, hydrosystem operations, and transportation—subsequently affected arrival timing at the Columbia River estuary, which in turn affected ocean survival. Although details of the migration models are unique to this system, the need to account for carryover effects and the correlation structure of climate drivers in multiple environments is widely shared by migratory species.

We used a stochastic, age-structured life-cycle model[37,38] with both density-dependent and density-independent climate effects. Survival of Chinook salmon (*O. tshawytscha*) was forced by environmental drivers, while future climate trends were based on ensemble projections from GCMs (Figs. 1 and 2 and Supplementary Data 1). We used a simulation framework to explore model assumptions and quantify different aspects of model uncertainty, including the functional form of the model, covariate selection, and life-stage-specific sensitivity.

We applied the model to eight populations within the Snake River spring/summer Chinook salmon Evolutionarily Significant Unit (ESU). This ESU migrates downstream from natal headwater streams in central Idaho, passing eight major hydroelectric projects on the Snake and Columbia Rivers to reach the northeastern Pacific Ocean (Fig. 3). After 1–4 years of ocean rearing, individuals return to freshwater and migrate upstream for a single spawning opportunity.

We simulated population time series for these exclusively wild populations, whose recent 15-year geometric mean spawner abundance ranges from 45 to near 600[39,40]. The model reconstructed historical population dynamics closely for most

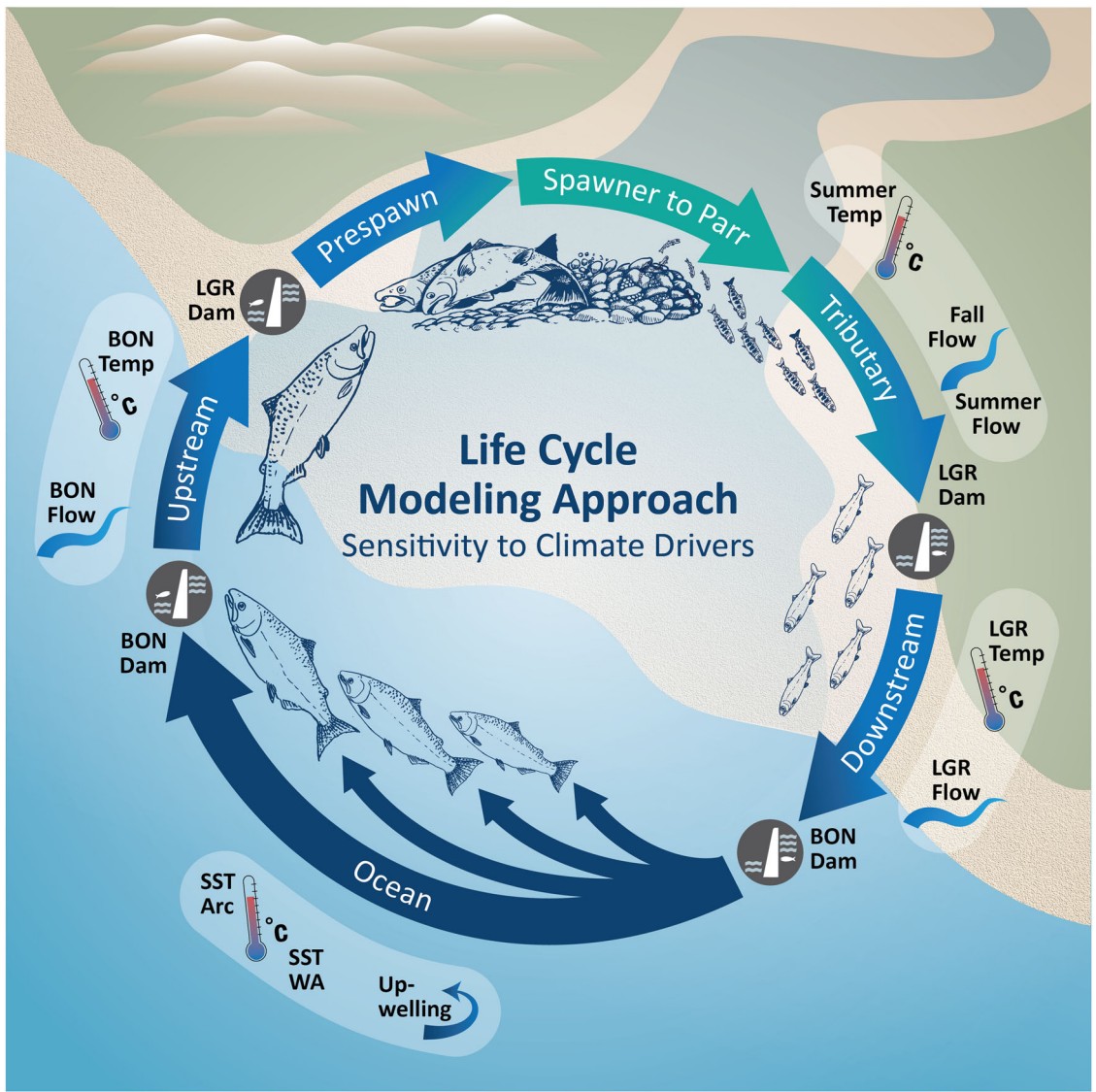

**Fig. 1 Life-cycle model diagram.** Diagram showing the life stages and climate drivers that influenced each stage. The freshwater stages include upstream migration and holding, spawning, rearing in tributaries, and migrating downstream. Migration routes pass through the Columbia River hydrosystem, bracketed by Lower Granite Dam (LGR), and Bonneville Dam (BON). Freshwater stages were influenced by stream flows (Flow) and temperatures (Temp). The marine stage, which has variable duration represented by multiple arrows, was influenced by ocean temperatures across the northeastern Pacific (SSTarc) and along the Washington coast (SSTwa). Other influences on ocean productivity are represented by Upwelling. Figure courtesy of Su Kim (NOAA Fisheries) and salmon illustrations by Blane Bellerud (NOAA Fisheries).

populations based on Kolmogorov–Smirnov diagnostics. This model has been relatively stable when confronted with new data over the last 10 years, a period during which it has been used operationally by the National Marine Fisheries Service[40]. Our updated version of the model is similar in general characteristics to that used by Crozier et al.[37]. However, the updated version imposes climate influences in more life stages and has stronger validation based on data from over one million individually tagged fish (Supplementary Methods and Supplementary Table 1).

We compared multiple alternative covariates in survival models, acknowledging that the best predictor might change over time and thus might track different temporal trajectories[25]. Notably, we have found that sea surface temperature (SST) is an important component of most relevant indices. This is not surprising, because SST reflects complex interactions between atmospheric forcing, wind strength, upwelling, and mixing of

ocean layers, all of which affect ecosystem productivity throughout the California Current[41]. Although the response of these processes to greenhouse gas forcing will vary, SST will likely continue to be an important indicator.

## Results

**Populations declined dramatically in a warming climate**. Chinook salmon migration timing did shift earlier in response to warmer freshwater conditions. However, the timing shift did not prevent large-scale population declines under either the RCP 4.5 or 8.5 scenarios. Population abundances diverged quickly under the two climate scenarios compared with baseline conditions, which were represented by a stable climate (Fig. 4). Recent population dynamics and historical population modeling are detailed in "Methods".

With a warming climate, deterministic declines inevitably lead to extinction unless some ecological, evolutionary, or climatic rescue

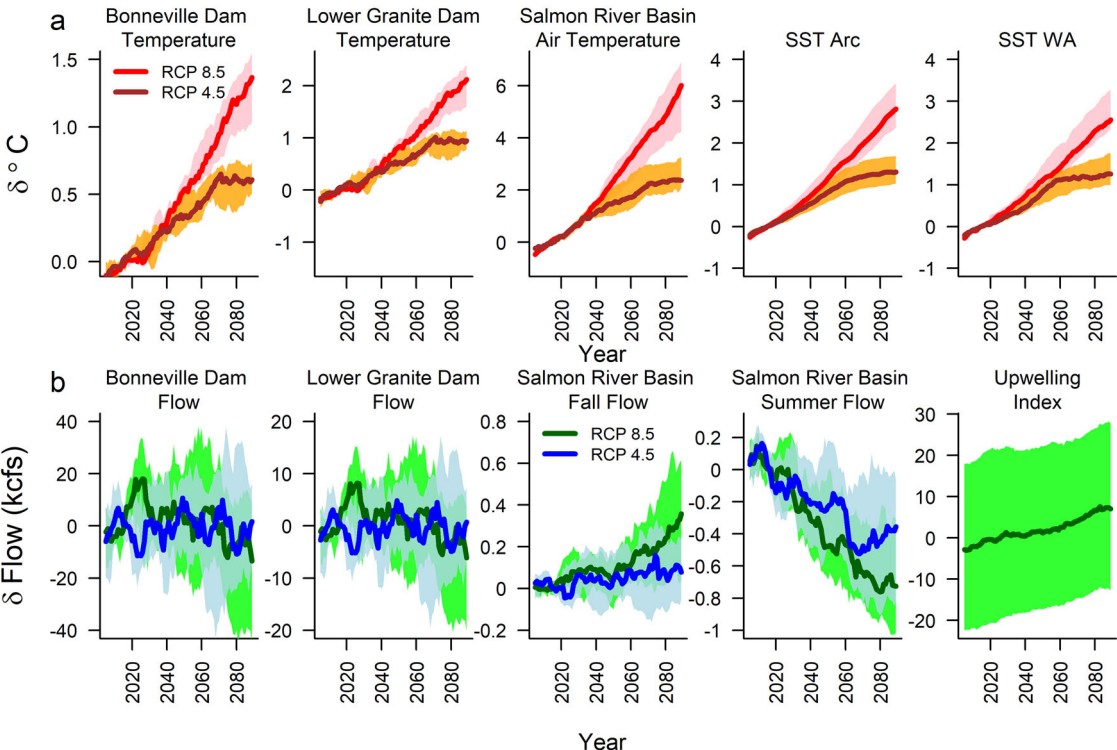

**Fig. 2 Trends from global climate model projections used in climate scenarios.** Global climate model projections are shown for each covariate that influenced population outcomes. Lines show median projections across climate models for two representative concentration pathways (RCPs), while shading shows the range across projections from the 25th to 75th quantiles. **a** Temperature covariates are shown in red for RCP 8.5 and brown for RCP 4.5, while **b** flow covariates are shown in green for RCP 8.5 and blue for RCP 4.5. For sources, see Supplementary Table 1. Note that the coastal upwelling index has different units from other covariates[112].

effect occurs[42]. Climate trajectories did level off in the second half of the 21st century under the RCP 4.5 scenario. This leveling reduced rates of population decline; however, most populations had already reached very low abundances at that point.

For practical purposes in salmon management, populations that have fewer than 50 spawners on average for 4 years in a row are considered to be at extremely high risk of extinction from chance fluctuations in abundance, depensatory processes, or the long-term consequences of lost genetic variation[43,44]. The evolutionary theory behind this threshold applies to isolated populations, and these populations are not truly isolated. So some small populations may be sustainable within a larger metapopulation. Nonetheless, when the majority of populations within an ESU pass this threshold, the ESU itself is at high risk. This ESU is already threatened with extinction because of historical declines[39]. Thus, although the 50-spawner threshold is somewhat arbitrary, it is a useful metric for demonstrating the severity of declines across all simulations. We assessed the first year (if any) in which a population in a given simulation fell below a quasi-extinction threshold of adult abundance (QET50). The QET50 is passed when the running mean of spawners, measured at the spawning stream, drops below 50 individuals over any 4-year period[45].

The proportion of all simulations in which a population dropped below QET50 increased dramatically under a warming climate compared with a detrended climate (Fig. 5). Under the representative concentration pathway (RCP) 8.5, ensemble-mean projection, even the largest populations (Bear Valley Creek and Secesh River) fell below QET50 in over 75% of simulations by 2060. Under RCP 4.5, the same milestone was passed a decade later.

**Similar results with alternative model covariates.** We conducted sensitivity analyses to characterize how extinction risk varied

depending on covariates included in the model, and the extent to which impacts at different life stages predicted population-level response.

Alternative marine and freshwater covariate models had relatively little effect on overall patterns (Fig. 6). Resulting from the high importance of SST in model comparisons, all three models included combinations of basin-wide and coastal SST indices, which drives much of the overall effect on survival. Freshwater models differed in whether fall (Models 1 and 3) or summer (Model 2) flows limited smolt productivity. We considered both because the data were consistent with both models. Populations fared slightly worse in summer flow models because projected decreases in summer precipitation, increases in evaporation, and reduced groundwater storage cause summer flows to decline (Fig. 2). In contrast, fall precipitation (and hence fall flow) is projected to either stay the same or increase (Fig. 2). Models that included stronger increases in upwelling (blue, Q75 compared with Q25 in Fig. 6) were more optimistic because upwelling is positively associated with salmon survival. Nonetheless, negative effects from SST still drove most populations extinct within the century (Fig. 6).

**Marine life stage most vulnerable to warming.** The impact of climate change on freshwater life stages was not always negative but rather depended on population-specific and model-specific sensitivity. In the headwaters, the fall flow model produced a net benefit to parr-to-smolt survival, whereas the summer flow model lowered survival in the juvenile rearing stage (red and blue vs. yellow boxes in Fig. 7), but produced no difference in effects at other life stages. Thus, if climate change affected only the early rearing period and no other stage, there could be a mix of responses across populations depending on local limiting factors

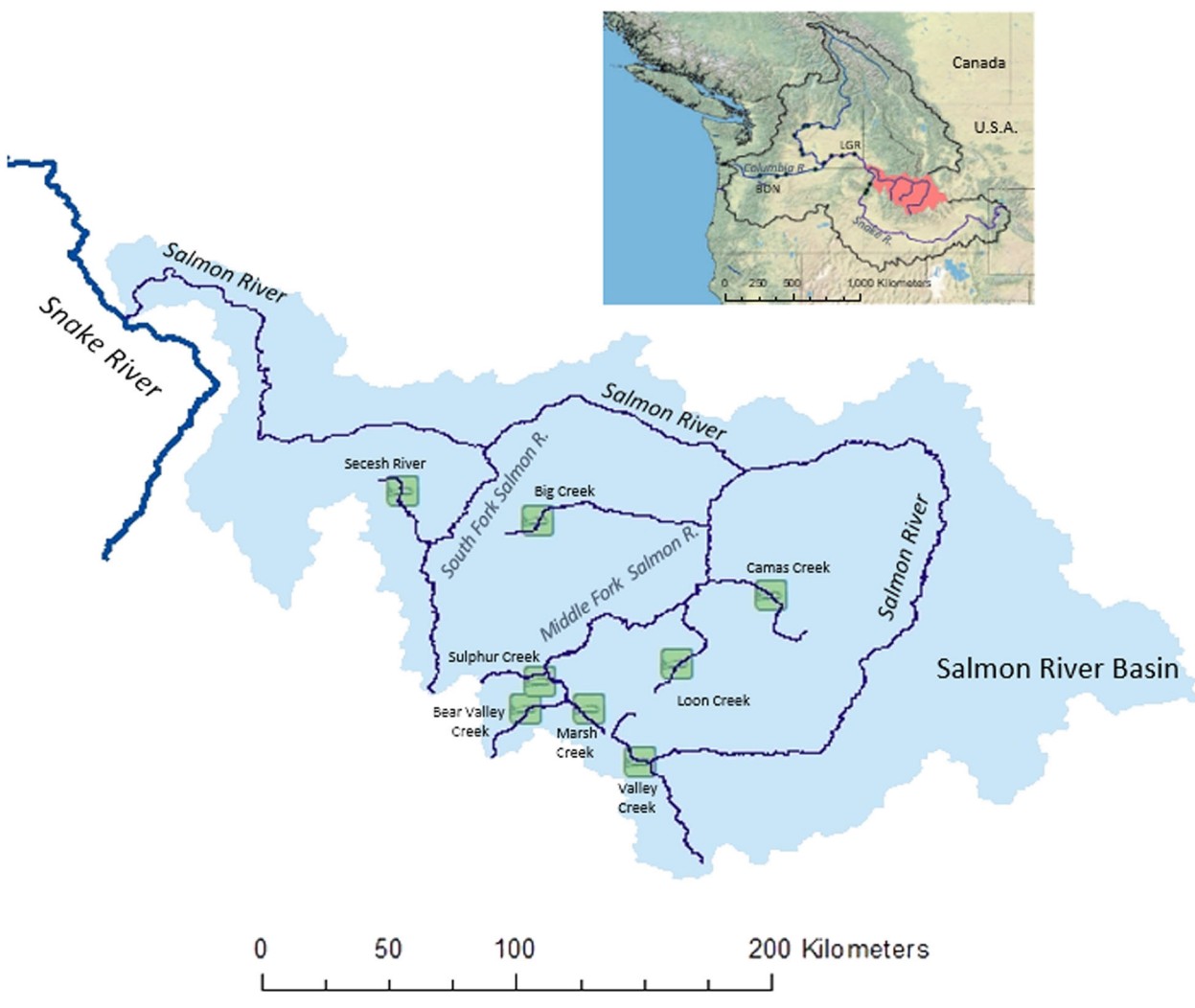

**Fig. 3 Map of study populations.** Chinook salmon populations included for modeling (green boxes in the lower map) are named by their natal stream in the Salmon River Basin. Insert above shows Columbia River Basin (black outline), with locations of the Salmon River Basin (red), Bonneville Dam (BON), and Lower Granite Dam (LGR).

such as summer vs. fall flow, similar to the conclusions of Crozier et al.[37].

Our model predicted that smolts would shift migration timing to arrive about 4.5 days earlier at Lower Granite Dam. This shift did indeed reduced temperature exposure in this life stage. Nonetheless, temperature effects on the juvenile migration still grew over time and reduced populations by about 18% on average from the 2020s to the 2060s.

Climate impacts were most dramatic in the marine stage, where survival was reduced by 83–90% (Fig. 7). This occurred despite the fact that smolts arrived at Bonneville Dam about 6.5 days earlier, indicating an earlier initiation of the marine stage, which generally improves marine survival[46].

Adult Chinook were also predicted to shift migration ~4 days earlier in response to warmer mainstem conditions with lower flows[47]. But again, this timing shift was not enough to prevent mortality from increased heat exposure. During the adult migration, populations that returned to spawning areas in summer (Secesh River and Valley Creek) were more affected by temperature than those that returned in spring, with net declines of up to 17% by the 2060 s. Still, the declines we found in the mainstem were relatively small because of the early adult run timing of Snake River spring/summer Chinook compared with other run types or species that migrate during peak temperatures.

## Discussion
Our analysis showed relative resilience in freshwater stages, with the dominant driver toward extinction being rising SST, which tracked a ~90% decline in survival in the marine life stage. This occurred despite an advance in smolt migration timing and other changes in hydrosystem management. The modeled carryover effects of changes in timing are likely to be adaptive, but inadequate as compensation for large declines in marine survival.

Our results indicate that as one symptom of a changing ocean, rising SST puts all of our study populations at high risk of extinction, despite actions within the hydrosystem to speed juvenile travel and increase in-river survival. Small populations had minimal demographic buffers against declining marine survival rates and quickly dropped below the quasi-extinction threshold in nearly all simulations (Fig. 5). Threats to our larger study populations caused even greater concern because they are the remaining strongholds, which provide genetic and demographic resilience for the spring/summer Chinook salmon ESU as a whole[48].

Although we have focused on details for certain populations, there is strong synchrony across all populations within this Chinook ESU[49] and more broadly across other salmon species throughout their southern distributions[50,51]. Such synchrony

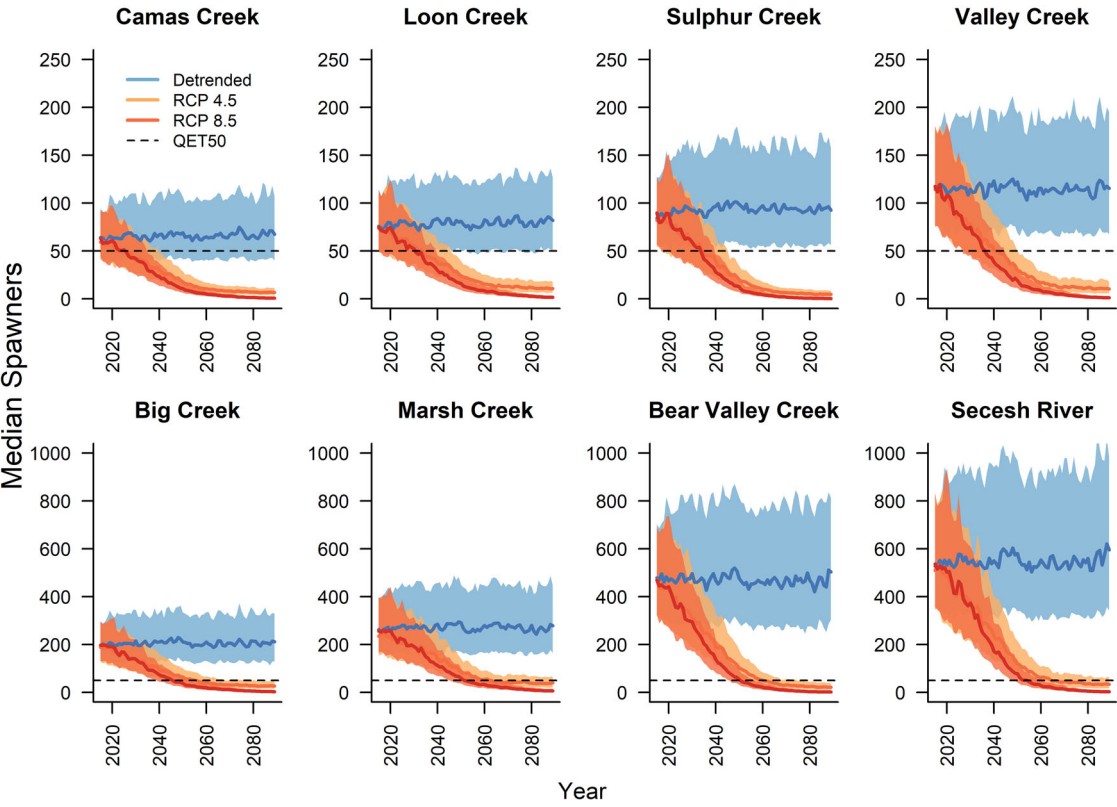

**Fig. 4 Population abundance over time by population.** Solid lines show the median while shading shows the interquartile range of population abundance across simulations for detrended (blue) vs. ensemble-mean global climate model projections for representative concentration pathways (RCP) 4.5 (orange) and 8.5 (red). Each panel shows a different population, ordered by present abundance. A reference abundance of 50 spawners, the quasi-extinction threshold (QET) is also shown (dashed horizontal line). Note the different y axes for small and large populations.

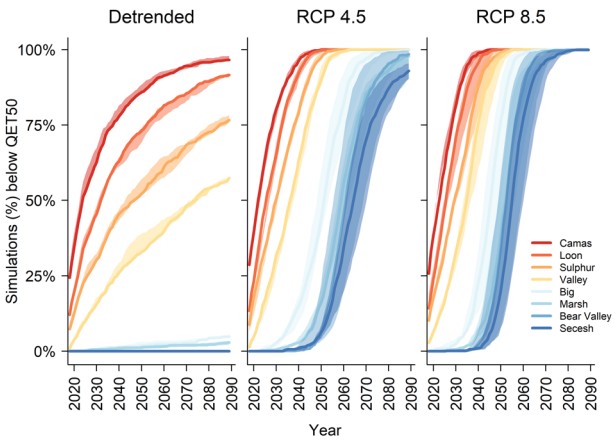

**Fig. 5 Percent of simulations in which individual populations reached the quasi-extinction threshold.** Time before most populations reached the quasi-extinction threshold (QET50) was longer in the detrended climate than in climate change scenarios RCP (representative concentration pathways) 4.5 and 8.5. For each climate scenario shown, solid lines reflect forcing by the median GCM projection, and shading reflects population responses across the interquartile range of projections.

suggests that the responses we found could represent a wide-spread phenomenon[52]. Salmon populations show coherence at many spatial scales, and synchrony has increased over time in both Pacific and Atlantic populations and in climate indicators[3,53,54].

Some Alaskan populations may fare better in response to ocean warming, but negative responses to warming in freshwater have been observed in other Alaskan populations, highlighting the benefits of examining the full life cycle[11,50,55]. And some studies have found similar declines in ocean survival over recent decades in southern and northern Chinook salmon[52]. Quantitative comparisons of combined climatic and anthropogenic influences, as well as more exploration of changing relationships among drivers, are needed to unravel the multiple pressures these populations face[4,56,57].

There are two main caveats to these modeled projections. First, the Northeast Pacific might not warm at the rate modeled, despite rising levels of atmospheric $CO_2$. Over the past century, global mean temperature has been a weaker determinant of SST than internal variability in the climate system, represented by strong changes in sea-level pressure and natural variability in ocean circulation[58]. How long this situation will continue is difficult to predict. Nonetheless, with the entire ocean warming at all depths[34], this signal will inevitably reach coastal waters.

A second possibility is that the Northeast Pacific will warm as modeled, but with some sort of ecological surprise[59] that will reverse the historical relationship between SST and salmon survival. Ocean temperature does not limit salmon through a direct physiological response, but rather through a combination of bottom-up and top-down trophic processes, which jointly regulate salmon growth and survival[60–62] and which explain the non-stationarity of statistical correlations[25]. Although warm conditions have been associated with lower-quality prey and more warm-water predators, it is possible that novel communities will arise with different responses to temperature, or that salmon will adapt to an altered food web in a positive manner. For example, salmon consumption rates could increase[63] with changes in the distribution of prey species[64]. Results from Model 3, in which marine survival was

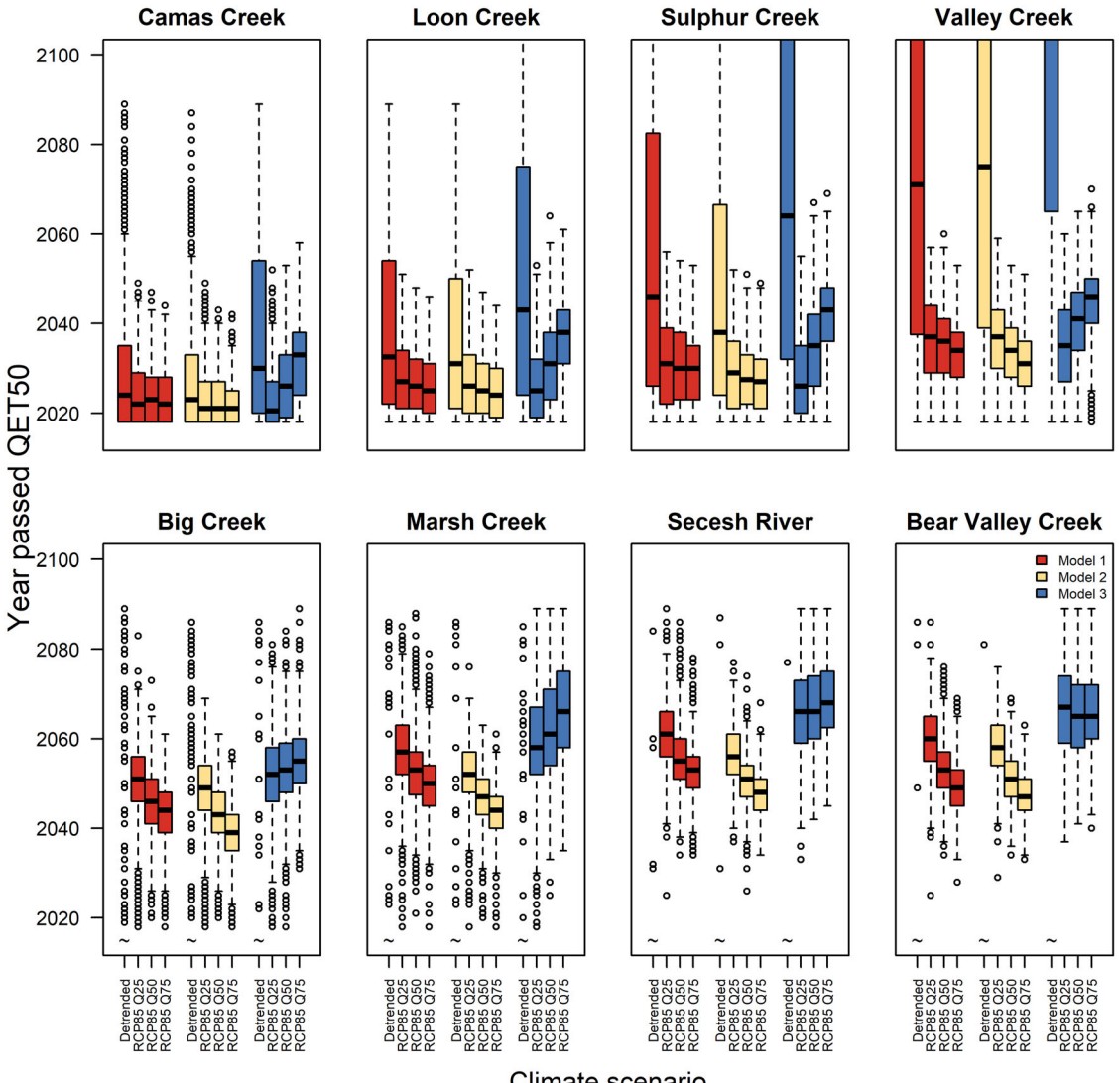

**Fig. 6 Comparison of alternative models in the year each population met the quasi-extinction criterion.** The year in which each population fell below the quasi-extinction threshold (QET50) for Model 1 (red), Model 2 (yellow), and Model 3 (blue). Model 1 covariates include sea surface temperature (SST), summer air temperature, and fall flow; Model 2 covariates include SST and summer flow; Model 3 covariates include SST, coastal upwelling, summer air temperature, and fall flow. Extinctions occurred earlier in scenarios with more warming (Q75) compared with less warming (Q25) in models that did not include upwelling (yellow and red) but later when more intense upwelling ameliorated SST effects (blue). However, in all models, extinctions occurred earlier in RCP 8.5 scenarios compared with a detrended climate. Note that larger populations (bottom row) often never dropped below QET50 in the detrended climate, indicated with ~. Boxes show the interquartile range across simulations, while whiskers extend to 1.5 times the interquartile range. Horizontal lines show the median values.

driven by upwelling, indicated that improved productivity in a warmer ocean could benefit salmon. Nonetheless, the benefit, in that case, was relatively small compared with overall negative effects. Further exploration using our model with a changing correlation structure over time could clarify additional possible trajectories and when they could be detected.

Other ecological surprises should also be considered, such as increases in competitors such as jellyfish[62] and Humboldt or market squid[65], either of which could reduce salmon survival in an altered ocean. Predators, which now concentrate on alternative prey, may change behavior when their preferred prey dwindle or shift distribution. Such behavioral change can either increase or decrease predation on salmon[66]. We recommend closely monitoring trophic interactions and salmon growth rates to detect such a possibility.

Finally, our model is conservative in that we have not accounted for any negative effects of ocean acidification. Declines of sensitive species could have a negative effect on salmon, especially salmon populations that rely more heavily on sensitive species[67]. We have assumed that ocean-stage salmon are relatively insensitive to pH, but if there are effects from ocean acidification, they will likely be negative[68,69].

Freshwater climate impacts in this study were also conservative in some respects. Specifically, we assumed linear responses to environmental variables, but actual physiological and ecological thresholds can create non-linear responses, meaning future temperature or flow impacts could have more severe effects. Furthermore, the primary covariate for juvenile survival in two of our models was fall flow, which was the covariate least sensitive to climate change[70]. Summer flow could also be (or become)

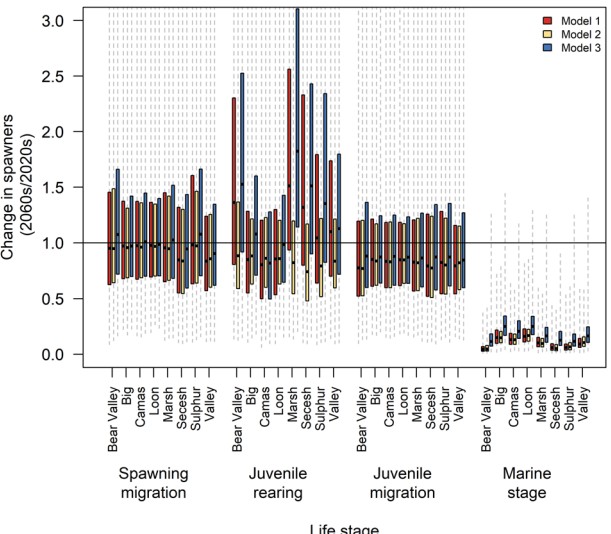

**Fig. 7 Sensitivity analysis of the impact of climate change on individual life stages.** Results from a three-by-four factorial experimental design examining relative loss in spawner abundance from 2020 to 2060 due to climate-change effects at four specific life stages. Net trends in population size are shown using the ratio of late abundance (2060–2069) compared with early abundance (2020–2029). Representative concentration pathway (RCP) 8.5 ensemble-mean climate trends were imposed at each life stage shown, with other life stages remaining under a detrended climate scenario. A value of 1.0 (horizontal line) indicates no change in spawner abundance. Boxes show the interquartile range across simulations, while the whiskers extend to the most extreme values of individual simulations. Horizontal lines show median values. Model 1 (red) covariates include sea surface temperature (SST), summer temperature, and fall flow; Model 2 (yellow) covariates include SST and summer flow; Model 3 (blue) covariates include SST, upwelling, summer temperature, and fall flow.

limiting (Model 2), and would cause a more negative response. More generally, the high-elevation, mostly-wilderness habitat of these populations is unusual for salmon in the region, and partially explains the relatively small effects of climate change on their freshwater life stages[71]. Other populations face more immediate impacts on freshwater productivity[16,72].

There is no easy way to completely mitigate for increasing SSTs that produce declines in marine survival of this magnitude. Additional changes to the juvenile transportation schedule or faster transit through the mainstem can still be explored, but improvements to survival through other mechanisms are needed. Ecosystem-based fisheries management attempts to consider the complex web of drivers for sustainability. These include both direct and indirect links to anthropogenic actions, as well as tracking of indicators in all sectors[73,74]. Conceptual models of factors that affect salmon marine survival show a large number of interacting processes[75,76]. Importantly, solutions involving management actions can be implemented in either the marine or freshwater realms.

In the marine realm, human activities can affect salmon survival through the harvest of salmon, their prey (sardine, anchovy, krill, juvenile rockfish, juvenile crab), predators (e.g., marine mammals), and competitors (e.g., hake, Pacific halibut, arrowtooth flounder). Changes in fisheries can consequently have positive or negative effects on salmon. Curtailed salmon harvest since the 1960s reduced one major source of direct mortality. Reduced harvest on top predators, however, led to rebounding marine mammal populations that now consume large amounts of salmon[61,77].

Sea-level rise combined with coastal development threatens complete loss of intertidal marshes in California and Oregon, as well as the majority of estuary habitat in Washington[78]. Such development will affect some salmon and their prey. Although the populations studied here generally spend relatively little time in the Columbia estuary, smaller, earlier-migrating individuals do utilize these estuary habitats[79]. As juvenile fish migrate earlier in the future, they might rely more on estuary habitat. Furthermore, other salmon populations depend heavily on estuary habitat for essential growth, and their success has been linked to estuary restoration actions[80].

Competition with hatchery fish interacts with climate effects to impact wild salmon in a complex manner[56] and is an area of active research. Reducing hatchery production could have positive benefits, especially for wild Alaskan salmon, which might then benefit from warmer conditions in some localities[11,55].

Other indirect anthropogenic effects on marine habitats are not well quantified and warrant more research. Increased awareness of the importance of forage fish for the entire food chain[81] has led to a ban on the development of new fisheries to exploit forage fish, demonstrating a proactive approach that should support salmon. But in sum, we lack key information on the full mechanistic basis of salmon marine survival, which limits the strength of end-to-end models for guiding management[82].

Efforts to mitigate carryover effects from freshwater that could affect marine survival in these populations have primarily focused on dams. Survival through Columbia and Snake River dams generally now meets recovery targets (>96%)[83], and cumulative mortality over 500 km of in-river migrating fish (~50%) is similar to that estimated for unregulated rivers of similar length (i.e., Fraser River[84]). However, slow travel time through slow-flowing reservoirs behind dams, combined with increased surface temperatures in these reservoirs[85], can potentially result in lower marine survival[86]. Mitigation efforts to increase smolt body size and advance migration timing could increase marine survival[46,87]. Restoration efforts in freshwater habitats, such as restoring floodplains, riparian planting to reduce stream temperature, reconnecting side-channel habitat[88], and supporting other natural processes in juvenile salmon rearing areas also enhance/restore freshwater salmon production.

Our models estimated strong density dependence at the parr-to-smolt stage, although it is not clear whether summer or winter habitat is constrained. Headwater areas are subjected to minimal anthropogenic impacts, but historical and current land use and irrigation withdrawals reduce habitat quality throughout the basin. Numerous actions have been identified in recovery plans and biological opinions that would improve rearing and migratory habitat in the mainstem Salmon River[40,89]. Our results suggest that at present, smolt carrying capacities are limited by flow more than temperature. Higher flows may create more habitat, improve connectivity, or decrease contact with predators. The predator community has also been affected by human impacts, from introduced sport fish (small-mouth bass and brook trout) to the creation of reservoir habitats extremely favorable to invasive fish species (e.g., American shad)[90].

Throughout salmon watersheds, improving and expanding access to rearing habitat should increase smolt abundance and body condition, resulting in improved population viability[91]. Intrinsic habitat potential is negatively correlated with present levels of disturbance, so restoring all critical habitat could yield substantial benefits. Specifically, the lower-elevation habitat that was historically highly productive has been preferentially lost. Improving individual fish growth by reducing contaminant loads[92], increasing floodplain habitat[91], and increasing habitat complexity, in general, could boost population productivity[93].

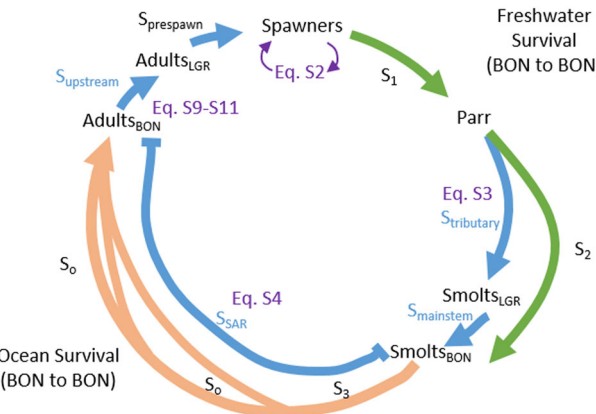

**Fig. 8 Diagram of life-cycle model.** Environmental covariate survival models that were fit directly to PIT-tag data are shown in blue ($S_{tributary}$, $S_{mainstem}$, $S_{SAR}$, $S_{upstream}$). Life stages and fitted transition parameters are shown in black, with associated equations for reference in the Supplementary Methods.

In conclusion, we have shown that prospects for saving this iconic keystone species in the conterminous U.S. are diminishing. Resilience to climate change depends on genetic and ecological diversity to adapt to environmental change. Many options have been considered over decades of dedication to salmon recovery, and improvements have been made. However, there are hard choices where human demands on land and water have come at the cost of wildlife.

The urgency is greater than ever to identify successful solutions at a large scale and implement known methods for improving survival. Management actions that open new habitat, improve productivity within existing habitat, or reduce mortality through direct or indirect effects in the ocean are desperately needed. We can find new ways to improve salmon habitats while maintaining other benefits for people, like reconnecting floodplains with rivers and natural marshes to recharge aquifers and mitigate flooding, storm surge, and channel erosion. Our modeling approach, which accounts for carryover effects across the life cycle, allows systematic exploration of alternative correlation structures among climate drivers and between climate drivers and ecological responses. This approach can provide a thorough accounting for uncertainty in climate projections and thus lay a path forward for evaluating the benefit of management actions protecting critical resources.

## Methods

**Data informing survival estimates**. Spawner age and abundance estimates were compiled through large-scale collaborative efforts between states, tribes, and coordinating bodies[94–96]. Stage-specific survival estimates were obtained from multiple sources[97–100], originating from tagging and detection records downloaded from PTAGIS.org, a regional collaborative database (Supplementary Table 2). We used detection records only of fish identified as wild from known population sources. No hatchery fish were released in any of the streams supporting populations included for this analysis. Environmental covariates used in the model included air temperature, streamflow and temperature, SST, and coastal upwelling (Supplementary Table 3).

For the smolt-to-adult analysis, we considered data for all out-migrating yearling spring/summer Chinook salmon tagged in the Snake River Basin detected from 2000 to 2015 at Bonneville Dam—the furthest downstream dam on the Columbia River. We marked a fish as having survived the marine stage if it was detected at Bonneville Dam (or farther upstream) as an adult. The data included (i) the last detection date at Bonneville Dam as juveniles, (ii) transportation classification (transported or in-river), and (iii) whether the fish was detected in the Columbia River as an adult. Fish were identified by Columbia Basin Research (CBR) as having migrated volitionally or having been transported by barge. All PIT-tag files are available on the CBR website (http://www.cbr.washington.edu/

dart/cs/data/nmfs_sar/). We excluded all fish with an unknown rearing type (i.e., hatchery versus wild), geographic regions with fewer than 200 individuals (over the 16 years), those fish released or tagged below the confluence of the Snake and Columbia Rivers, and fish that returned to spawn in less than 1 year. In addition, we excluded fish that passed Bonneville Dam prior to April 9th or later than July 8th (<0.14% of the total observations), because there was insufficient data to inform the temporal autocorrelation at the margins of the migration period. In total, there were 122,415 wild individuals for the SAR analysis.

**Life-cycle model structure**. We employed a stochastic, age-structured model modified from previous life-cycle models[37,38,101]. The model as depicted in Fig. 8 has five annual time-steps, based on the typical maximum 5-year generation time of Snake River spring/summer Chinook salmon. These time-steps correspond approximately to life-stage transitions.

The first time-step ("spawner to parr" stage) spans spawning (August/September), incubation, and early parr rearing. Survival through the second timestep ($S_2$) includes both tributary rearing ($S_{tributary}$) from summer (July or August) to the following spring when migrating juveniles pass Lower Granite Dam (April/May), and continue ($S_{mainstem}$) through the Snake and Columbia River hydrosystem.

The third timestep includes ocean entry (May/June) and the first winter and spring in the ocean. Some Chinook salmon return to spawn in their third year (jacks), but most females and mature adult males stay in the ocean for one or two more years, during which ocean survival is represented as $S_o$. The number of fish in the ocean each year is a latent variable, fit by detections of survivors when they re-enter freshwater ("smolt to adult return," or $S_{sar}$). Upstream migration survival through the hydrosystem ($S_{upstream}$) and from Lower Granite Dam to spawning areas ($S_{prespawn}$) are captured in the fourth or fifth timestep. Older females tend to lay more eggs, which is reflected in the fecundity parameter, $F_5$.

To calculate "effective spawners," we combined different age classes that returned to spawn in the same year as the weighted sum of three (weight = 0), 4- (weight = 1), and 5-year-old fish (weight = $F_5$). This sum was estimated by an expansion of the number of redds (nests) counted during spawning surveys[45].

**Statistics and reproducibility**. We fit the life-cycle model in two steps. First, we fit individual life-stage relationships with covariates using a variety of methods for different life stages to generate a posterior distribution for stage-specific parameter estimates. In addition to these parameters, the life-cycle model introduced some additional parameters that could not be directly fit to the data. We, therefore, conducted a second step to calibrate the life cycle model using the Approximate Bayesian Computing approach[102,103]. All analyses were conducted in RStudio (ver. 1.3.1073) using R (ver. 3.6.2)[104].

In the calibration step, we simulated population time series under recent climatic conditions using the life-cycle model. We compared the resulting spawner and smolt time series to those observed and ranked parameter sets by deviance from a Kolmogorov–Smirnov test. We selected the top 0.2% of 500,000 sets of parameter combinations (Supplementary Fig. 1). All resulting parameter sets met the criterion of producing a spawner distribution that was not statistically different from the observation dataset (P value > 0.05 from Kolmogorov–Smirnov test). This process maintained the appropriate correlation structure within each parameter set.

*Spawner to parr ($S_1$) and parr to smolt ($S_{tributary}$) stages*. We fit adult recruits per spawner for eight populations in a hierarchical Bayesian framework using multiple likelihood equations. These equations reflected stages that could be compared directly with data. Briefly, we fit a two-stage Gompertz function[105] (see Supplementary Methods) to solve the two stages simultaneously, combined with independently estimated survival rates for later stages. Individual population coefficients (productivity and capacity parameters for both stages as well as coefficients for temperature and flow) were assumed to be random samples from an underlying normal distribution[106] (priors shown in Supplementary Table 4).

To determine the best environmental covariates, we compared the estimated predictive error of alternative models using a leave-one-out, cross-validation method for Bayesian models using the LOO package[107]. Because of correlations among the climate variables tested, multiple models had similar support from the data (Supplementary Table 5). Our primary objective was to identify divergence in potential responses to climate change. Therefore, we selected two models with covariates that showed differing trajectories with climate change (Fig. 2). Model 1 included summer air temperature and fall streamflow, while model 2 included summer streamflow only.

*Juvenile survival through the Columbia River hydrosytem ($S_{mainstem}$)*. We estimated juvenile survival from Lower Granite to Bonneville and arrival day at Bonneville Dam using the COMPASS model[108]. In the COMPASS model, equations for each dam and riverine reach used measured hourly flow, temperature, and spill to predict fish survival and migration rate. Dams had additional equations to determine the proportion of fish that passed using the spillway, turbine, or bypass system as a passage route. The COMPASS model also tracked the proportion of fish loaded into barges for transport and release downstream from the hydropower system.

*Smolt-to-adult return (S_sar).* We used a mixed-effects logistic regression model to determine the effects of ocean entry date and environmental covariates on the probability that an individual fish would return as an adult to Bonneville Dam. The model[109] included random effects of day and day-by-year interaction, which followed an auto-regressive process. Using Akaike Information Criterion, we selected variables with high importance based on model weights, which included a large-scale measure of SST (SSTarc) and a more local, coastal measure of SST (SSTwa), as well as spring upwelling. We applied separate models for fish that had migrated through the mainstem river vs. those that had been transported (Supplementary Table 6).

*Adult upstream survival (S_upstream).* For the adult upstream survival model, we used generalized additive mixed models (GAMMs) to evaluate the effects of both anthropogenic and environmental covariates on spring/summer Chinook salmon survival[47]. To run the model in simulation mode, all the non-environmental covariates (fisheries catch, proportion transported as juveniles) had distributions similar to those during the baseline period of 2004–2016. Survival from the hydrosystem to spawning streams ($S_{prespawn}$), was held constant due to lack of appropriate data with which to fit a relationship.

*Calibration step.* The model included a set of maturation parameters that could not be estimated directly from the data. Therefore, we treated them as tuning parameters for the life-cycle model as a whole. These parameters partitioned total smolt-to-adult survival (SAR) into age-specific survival rates (S) and propensity to return at a given age (b) as follows:

$$S_{SAR} = \frac{b_3 \times N_3 + b_4 \times N_4 + N_5}{N_2} \tag{1}$$

where $b_3$ denotes the proportion of all 3-year-old fish ($N_3$) that returned as jacks, $b_4$ is the proportion of all 4-year-olds ($N_4$) that returned to spawn, and $N_5$ is all 5-year-old fish. We calculated the number of fish in each age class as follows:

$$N_3 = S_3 \times N_2 \tag{2}$$

$$N_4 = (1 - b_3) \times N_3 \times S_0 \tag{3}$$

$$N_5 = (1 - b_4) \times N_4 \times S_0 \tag{4}$$

where $S_3$ denotes the survival rate of all smolts ($N_2$) over the first winter in the ocean and $S_0$ is the survival rate over subsequent years at sea. Fish that stay in the ocean longer have additional mortality, but there is a fecundity advantage (F) for older spawners because larger fish have higher fecundity. In the model, the effective number of spawners reflects the age distribution of female spawners, which return as either 4- or 5-year olds, with the latter having a fecundity advantage. A percentage of fish returned as 6-year olds, but it was so small that we pooled them with the 5-year olds.

We fit the tuning parameters ($S_0$, $b_3$, $b_4$, and $F_5$) using a modified Approximate Bayesian Computing approach[102,103]. We applied this method by first generating a prior distribution for each parameter. The priors for parameters that range between 0 and 1 ($S_0$, $b_3$, $b_4$) had a beta distribution centered on the mean and included the full range used by Zabel et al.[38] (Table 1). For $F_5$, we used the normal distribution with mean from Kareiva et al.[98] with sd = 0.1. These values were randomly combined with the freshwater parameter sets to create 500,000 combinations of all ten parameters.

We ran the life-cycle model to tune each population using each of these parameter sets in a different simulation. The life-cycle model was forced by historical meteorological conditions from 2000 to 2015 (Supplementary Table 3). For juvenile mainstem survival, we used COMPASS reconstructions of juvenile migration that incorporated actual river management practices. Note that historical river management was extremely variable over the juvenile migration stages and not comparable to the conditions we projected in climate simulations. In particular, transportation rates were higher, the spill was lower, and dam-passage survival was lower than in the corresponding actions proposed by the U.S. Army Corps of Engineers et al.[110] in their environmental impact statement. We ran the SAR model in retrospective mode, which used the fitted estimates of historical random effects and observation error. We simulated other life stages in the calibration as we would in future projections.

We ran 500,000 iterations with different parameter combinations through the historical reconstruction. We selected this number to produce at least 1000 sets of parameter values for each population that produced spawner distributions similar to those observed (Fig. 9). Parameter values from the top 0.2% of these parameter sets are shown compared with their prior distributions in Supplementary Fig. 1. The resulting models produced Kolmogorov–Smirnov P values from 0.16 to 0.76, each of which rejected the null hypothesis that modeled and observed time series came from different distributions ($P < 0.05$).

## Climate scenarios

*Detrended climate simulations.* In creating future climate scenarios, our aim was to maintain the statistical properties of the environmental data driving survival in various life stages. Large-scale oceanic and atmospheric drivers affect marine and freshwater environments simultaneously. To account for this, we estimated an unstructured covariance matrix for all the freshwater and marine environmental covariates using the *Template Model Builder* (TBM) libraries for R[111]. We used a multivariate state-space model

$$\mathbf{x}_t \sim \text{MVN}(\rho^x \mathbf{x}_{t-1}, \mathbf{Q}) \tag{5}$$

$$\mathbf{y}_t \sim \text{N}(\mathbf{x}_t, 0.001) \tag{6}$$

where $x_t$ was a vector of environmental processes at time $t$, $\rho^x$ was the correlation between vectors from successive time-steps, and Q was an unstructured covariance matrix. For $n$ covariates, the number of estimated correlation coefficients in the unstructured covariance matrix was equal to $0.5 \times n \times (n - 1)$. Observation error for the observation model was fixed at 0.001, essentially implying that covariate data were measured without error.

**Climate change scenarios.** Our objective was to specifically explore how the relationships among various climate drivers interacted with population dynamics to shape population trajectories. To do this, we first simulated 1000 time series of 75 years of detrended environmental conditions that followed the model-fits of the covariance relationships in the historical record (Supplementary Fig. 2), although different matrices were generated for different models.

Second, we added offsets to these detrended simulations according to GCM projections. Each offset (trend) consisted of a single time series added to all 1,000 detrended simulations. Then, for each climate scenario, we input the resulting combinations of detrended-plus-offset environmental conditions into the life-cycle model as forcing factors.

For each model/population/climate scenario, we ran 1000 iterations. Climate trends were created from the median (GCM_50) and interquartile range across 10–80 GCM projections, depending on the covariate (Supplementary Table 1). We tracked geometric mean population abundance in 10-year intervals. We also assessed the first year (if any) in which a population in a given simulation fell below a quasi-extinction threshold of adult abundance (QET50). The QET50 was passed when the running mean of spawners, measured at the spawning stream, dropped below 50 individuals in any 4-year period[45].

We explored projected climate trends from two carbon emissions scenarios, representation concentration pathway (RCP) 4.5 and 8.5[112,113]. For marine variables, we used time-series output from GCMs directly. For freshwater variables, we used output that had been statistically downscaled and then processed through hydrological and stream temperature models[70,114,115]. For each variable, 10–80 time series per emissions scenario were available (Supplementary Table 1).

We used four steps to extract the relevant trends from GCMs. First, we calculated the variable mean (e.g., the average for a given month) over a baseline period centered on 2015 (2005–2025) within each time series individually. Second, we calculated anomalies from the 2005–2025 baseline period within each time series. Third, we created a 20-year running mean of anomalies for each time series. We smoothed each of these individual time series using a 20-year running mean to reduce interannual variation that was already accounted for by the TMB model. Fourth, we calculated the 25th, 50th, and 75th quantiles across all time series for each covariate in each year from the smoothed anomalies.

We repeated this process for each climate scenario, producing the trends shown in Fig. 2. These trends represented the spread across climate models of the low, medium, and high rates of change in each covariate. The smoothed trends were nearly linear because decadal and sub-decadal variability averaged out across models for most variables. However, decadal patterns were still apparent in the smoothed trends for flow, due to its very high natural variability as well as the few GCMs driving flow projections.

Although temperature and flow are physically linked (air temperatures are input into the hydrological models, and both air temperature and flow are input into stream temperature models) we found the trends for temperature were much more linear than those for flow. Thus, there was no consistent relationship between them (i.e., the rank order of temperature projections was not strongly correlated with the rank order of flow projections). We, therefore, treated the temperature and flow offsets as independent, and retained decadal variation in flow, which was low relative to annual variation. Exploration of alternative methods indicated this choice did not influence population responses.

Thus, in summary, we added each climate trend to the raw simulation of a detrended climate produced by the covariance model. All new time series retained the autocorrelation, variance, and covariance of the historical environment with forcing from greenhouse gas emissions added.

**Life-cycle model simulations.** Running the model in simulation mode was relatively straightforward for stages that had been fitted on an annual timestep ($S_1$ and $S_{tributary}$) because they only required an annual input of environmental conditions. However, submodels for upstream and downstream survival required daily environmental inputs. We, therefore, ran a separate step to produce survival estimates from numerous representations of daily time series, which were then grouped by annual spring mean temperature and flow. For each annual timestep, we sampled randomly from the appropriate disaggregated submodel output (Supplementary Methods).

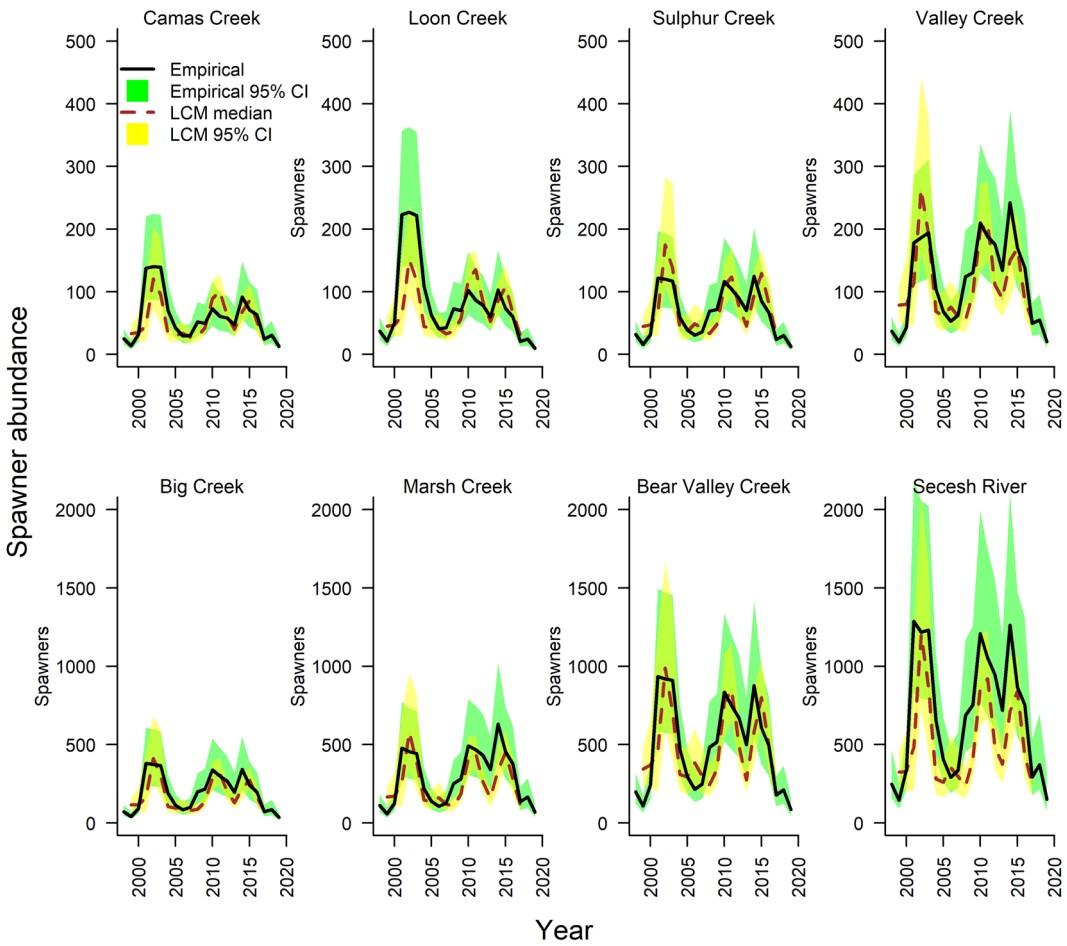

**Fig. 9 Historical time series of spawner abundance.** Heavy lines (black) show empirical estimates modeled from redd-count data for population status reviews, using methods described in ref.[39]. Broken lines (red) show median abundance from life-cycle model (LCM) simulations. The empirical estimates account for observation error and include 95% confidence limits, shown in green, while the 95% bounds from the life-cycle model are shown in yellow.

We ran the life-cycle model from 2015 to 2089, applying climate trends to the detrended climate for each environmental covariate on an annual timestep. We repeated 1000 iterations for each population, per model, and climate scenario. Our first results stemmed from a single combination of freshwater and marine covariates (Model 1: summer temperature + fall flow for freshwater, SSTarc in winter for transported fish, and SSTarc in winter + SSTwa in summer for in-river fish to show the full-time series of population response to climate scenarios (Figs. 4 and 5).

**Sensitivity analyses**. In our first sensitivity analysis, we compared population outcomes from Model 1 with a different freshwater covariate model (Model 2: summer flow + Model 1 marine covariates). Finally, we exchanged the marine covariates in Model 3: summer temperature + fall flow for freshwater, SSTwa in summer for transported fish, and SSTarc in spring + spring upwelling for in-river fish (Fig. 6). Thus, we explored all combinations of the top two models for freshwater and marine stages, respectively.

In our second sensitivity analysis, we applied climate trends for the ensemble mean of RCP 8.5 to one life stage, while the other life stages experienced a detrended climate (Fig. 7). We cycled through survival for parr-to-smolt stage ($S_{tributary}$), downstream migration ($S_{mainstem}$), smolt-to-adult return ratio ($S_{sar}$), and upstream spawning migration ($S_{upstream}$). In each case, we reported the extent of population decline as the ratio of geometric mean population size in 2060–2069 divided by mean abundance in 2020–2029. We ran 1000 simulations per life stage and calculated the mean change in abundance for each population.

**Reporting summary**. Further information on research design is available in the Nature Research Reporting Summary linked to this article.

## Data availability

All of the raw PIT-tag data used in this analysis are publicly available for download from PIT Tag Information System (www.PTAGIS.org). All survival estimates made from the raw data are published in other sources, listed in Supplementary Table 2. Redd counts are

available upon request from the Idaho Department of Fish and Game (IDFG.idaho.gov) for all but one population, and from the Nez Perce Tribe for the Secesh River population (https://www.cbfish.org/Document.mvc/Viewer/P165414). Population estimates from redd counts were published by Ford, et al.[39] and are available at https://www.webapps.nwfsc.noaa.gov/apex/f?p=261:HOME::::::#. Environmental data are all publicly available from sources cited in Supplementary Tables 1 and 3. The source data underlying Figs. 2, 4–7, and 9 are provided as Supplementary Data 1.

## Code availability

Model code is available on GitHub (lisa439/LCM).

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

## Acknowledgements

This work would not have been possible without the enormous effort to PIT-tag fish, design, build and maintain detection facilities, conduct redd counts, and manage the various databases. State (Idaho, Oregon, and Washington), federal (U.S. Fish and Wildlife and NOAA Fisheries), and tribal entities (especially Nez Perce) PIT-tagged most of the fish used in this study. Idaho Department of Fish and Game and the Nez Perce Tribe conducted most of the redd counts, and Mari Brick (NOAA Fisheries) assisted in managing the spawner database. We also thank Steve Achord, Gordon Axel, and Jesse Lamb, from the Wild Fish Monitoring Program at NOAA Fisheries for long years of tagging wild fish in natal river reaches. Susannah Iltus (Columbia Basin Research) assembled PIT-tag data according to juvenile transportation history. We also thank Jeff Jorgensen, Jennifer Gosselin, and Eric Ward for reviewing drafts of the paper.

## Author contributions

L.G.C. contributed to the conception, design, analysis, and interpretation of data and writing of the paper. B.J.B. and R.W.Z. contributed to design and interpretation of the work and manuscript revisions; B.E.C. contributed to the analysis, developing new methods for the marine survival stage, and writing. D.L.W. led data acquisition from PTAGIS, design, and analysis of the juvenile migration stage.

## Competing interests

The authors declare no competing interests.
