## [Peer Review File · Communications Biology]

Reviewers' comments:

Reviewer #1 (Remarks to the Author):

This manuscript describes an effort to use a state-of-the-art life-cycle model to quantify the risks of extinction for 8 populations of Chinook salmon in the Pacific Northwest of the USA. The model relies heavily on an extraordinary dataset of individually tagged fish to estimate various functional relationships that describe the processes that affect survival throughout the life-cycle of this species. The model is used to simulate the potential effects of climate change on the viability of these 8 populations. The primary conclusion reached by these efforts is that climate warming is likely to doom this group of populations and that the primary survival bottleneck is during the fish's first year of life in the sea. While the model development is commendable and the data used are impressive, the conclusions that are drawn are not particularly new and have been appreciated for some time. Thus, it is not really clear what the main contribution from this paper actually is.

My concerns for the current manuscript are given below.

1) The manuscript attempts to grasp at jargon to increase its general appeal, but these attempts are often a bit off-target. For example, there has been a lot of interest in 'non-stationarity' in ecology and in fisheries science recently and this manuscript attempts to link in to these interests. However, while this term does refer to a changing variable (in the strictest statistical sense), use of this term in ecology typically is referring to changes in a relationship between variables (i.e. the relationship or correlation among variables is non-stationary). Thus, why call climate change 'stationary' or 'non-stationary'? We have known for decades that climate is non-stationary; what is interesting is that the relationships between climate conditions and ecological processes are non-stationary. Thus, use of non-stationary/stationary in this manuscript is somewhat distracting. While this is just a semantic issue, I don't think the paper benefits from the current use of these terms.

2) Similarly, the manuscript refers to 'aggressive' warming scenarios. This also seems misplaced. Typically we refer to aggressive scenarios of curtailing carbon emissions (i.e. it's not the climate that is aggressive, it's the policy actions to reduce emissions that are aggressive).

3) The key results are expressed as the time to quasi-extinction for each of these populations. In general, I do not think this is the best way to present the key results. When a modeled population goes extinct in these simulations is based on arbitrary population thresholds that probably don't apply well when populations are reduced to very low numbers where stochastic processes are more likely to ultimately determine whether they go extinct or not. Thus, the primary results of these simulations would be more useful if the population growth rate was the response variable used to explore the consequences of different climate scenarios (i.e. the posterior distribution of lambda). The result could then be focused on how much of the posterior distribution was <1, thereby leading to population decline, etc. Time to extinction is too arbitrary and too 'loaded' a variable that is easily misinterpreted that it shouldn't be used in these types of analyses.

4) Line 24, 'species' should be 'populations'

5) ~ Line 121, it is not clear what the goals of the paper are. Is it to construct this model? The paper would benefit from some clear statements about what the primary objectives of this exercise are.

6) It is surprising that the model suggests that the populations are essentially at no risk of going extinct given the 'stationary' climate scenario. This does not seem realistic given the current status of these populations.

7) I think the 'Caveats' section should be part of the Discussion of the paper.

8) The Discussion highlights the weakness of the paper. Here the text basically runs through what is known about different processes or conditions that affect salmon survival. Nothing is particularly novel in this summary, nor does it highlight the main contributions of this specific modeling effort. Thus, it reinforces the lack of clarity about what this specific research activity contributes to our general understanding of Chinook salmon ecology and conservation.

Reviewer #2 (Remarks to the Author):

I found this paper to be exceptionally well written and the topic to be of broad international interest as a study of the impact of climate on a commercially valued species that is now at risk. This work explores the population dynamic of 8 populations of Chinook salmon in the Snake River watershed. This paper uses a life cycle model and two climate change scenarios to model the potential population trajectories of these populations. It effectively tells the risks of changing climate, particularly at sea, to all these populations but especially the smaller ones. It is sobering in its assessment of the viability of these populations- not a happy story. However, it effectively looks at the fact that these fish persist in highly altered systems and actions in freshwater might mitigate or at least slow population declines and localized extinction. These findings are novel for the populations of interest and findings (and solutions) will they be of interest to not only the fisheries biology community but ecologists and also to the public. This work is a tangible story of the impact of change upon a species that captures the imagination of a broad spectrum of society. The species also has rich cultural importance to first nations in this watershed. These conclusions are supported by appropriate modeling and add to not only Pacific salmon literature but will be of great interest to Atlantic salmon conservationists. The sea-run nature of this species also will inform understanding of challenges in freshwater and marine systems. I found this paper to be extremely convincing and the conclusions are strengthened by a rice use of citations and the addition of allied related manuscripts on the stream temperature model and wild/hatchery marine survival. This work is cutting edge and current. I firmly believe that this paper will influence thinking in the field due to its clarity of message and exceptional graphical presentation of results. I am generally familiar with salmon life cycle models and the overall use was both appropriate and added novel methods. I noted in one part of the paper where I am less familiar with one of the statistical analyses used. I would defer to you and others there. The open nature of the models, data and coding would allow not only the researchers to reproduce the work but add on to a toolkit of models that could be used for other species and habitats.

I have included comments in the margin of the manuscript as well as made some direct in-text suggestions to improve clarity. This was a pleasure to read.

John F. Kocik

Reviewer #3 (Remarks to the Author):

In this paper the authors apply a stage-based life history model to 8 populations of Chinook salmon in the Columbia River/Snake River basins and show strong associations between warming (particularly warming SST) during the marine stage of the life history and probability of population extinction/extirpation. I accepted the review with considerable excitement of seeing something truly new, insightful, or transformative. Unfortunately I was underwhelmed, not by the statistical rigour, but rather with the interpretation of the results. The claim is essentially that warming is bad and that smolts need to survive the ocean better for populations to avoid extinction. The authors all but said, and perhaps should have, that this analysis provides strong evidence that these populations are doomed and that restoration/conservation is a fools-errand (that would indeed have been provocative at least).

What I was hoping to see more of was a more holistic linkage between different stages of the life history and a quantitative appreciation that what happens in freshwater may lead individuals down trajectories that result in the ocean life history being the proximate stage of mortality. I was hoping and expecting to see discussion that warming SSTs may be detrimental to southern Chinook populations, but Alaska populations may fare better during warming temperatures (and indeed the authors did not cite the obvious paper to suggest so).

So in the end I am left wanting the authors to make a better case for novelty and insights that can be gleaned by this very complex modelling exercise that goes beyond what is already firmly established.

Although the reference section is extensive, I do suggest the authors incorporate information from populations beyond their focal range to broaden the discussion of SSTs and to also contrast their work to other very similar approaches.

Cunningham, Curry J., Peter AH Westley, and Milo D. Adkison. "Signals of large scale climate drivers, hatchery enhancement, and marine factors in Yukon River Chinook salmon survival revealed with a Bayesian life history model." *Global change biology* 24.9 (2018): 4399-4416.

And a new paper just out led by Leslie Jones (open access)
<https://onlinelibrary.wiley.com/doi/full/10.1111/gcb.15155>

Iconic species in peril: Chinook salmon face climate change across multiple life stages

Crozier, L.C.*, Burke, B. J., Chasco, B.E., Widener, D. L, Zabel, R.W.

Fish Ecology Division, Northwest Fisheries Science Center, National Marine Fisheries Service, National Oceanic and Atmospheric Administration, 2725 Montlake Blvd E., Seattle, WA 98112, USA

Contact information:

- [*Lisa.Crozier@noaa.gov](mailto:Lisa.Crozier@noaa.gov)
- Brian.Burke@noaa.gov
- Brandon.Chasco@noaa.gov
- Daniel.Widener@noaa.gov
- Rich.Zabel@noaa.gov

Keywords

Columbia River Basin, Snake River salmon, spring/summer Chinook salmon, conservation, biological impacts of climate change, ecological modelling

**Abstract:**

Widespread declines in salmon have tracked recent climate changes. But managers still
lack quantitative projections of population viability in response to future climate change. We
addressed this gap by assembling a vast database of survival and other data for eight wild
populations of threatened Chinook salmon. We evaluated climate impacts at all life stages, and
modeled future population trajectories forced by global climate model projections. Populations
rapidly declined in response to increasing sea surface temperatures and other factors across a
wide range of model assumptions and climate scenarios. Strong density dependence in
freshwater habitat limits the number of salmon that survive early life stages, suggesting a
potential target for conservation effort. Other potential solutions require a better understanding of
the factors that limit survival at sea. We conclude that dramatic increases in the number or
survival of smolts are needed to overcome the negative impacts of climate change for this
threatened species.

Introduction

The worldwide decline of wild salmon ¹⁻³ has negatively affected fisheries, cultural
heritage for indigenous tribes ⁴, and other marine species, including endangered Southern
Resident Killer Whales ⁵. Currently the majority of Atlantic and Pacific salmon and steelhead in
the conterminous U.S. are threatened with extinction ⁶. Overfishing, migration barriers, water
diversions, habitat loss, salmon farms and hatcheries drove much of the decline, as well as
previous regime shifts in marine ecosystems ⁷. Climate change has increasingly become an
additional threat, as remnant small populations in overly simplified habitats may lack the
adaptive capacity necessary to cope with it ^{8,9}. Quantifying the impacts of climate change is a
high priority for salmon management, driven by legal requirements ¹⁰ and national guidelines on
science strategy ¹¹. Despite these needs and despite the vulnerability of Atlantic and Pacific
salmon to climate change through freshwater and marine forces ^{12,13}, quantifying these threats
throughout their life cycles with future climate projections has been problematic.

Previous population models that have used global climate model (GCM) projections have
focused on freshwater life stages only (e.g., stream temperature, winter flooding, and drought) ¹⁴⁻
¹⁶. GCM projections related to marine survival, on the other hand, have primarily been used to
inform niche-based models that forecast future habitat for salmon generally rather than for
specific populations ^{17,18}. Although marine climate indices have been tightly linked to survival
(e.g., Pacific Decadal Oscillation (PDO), ¹⁹Atlantic Multidecadal Oscillation (AMO)), many
cannot readily be used in projections of future climate because GCMs have low confidence for
these metrics ²⁰. Moreover, ocean stratification may change their characteristics, leaving
biological impacts uncertain ²¹. Our analysis was focused on climate drivers with more reliable

GCM performance that are also closely correlated with fish survival to produce a more robust
analysis of the impacts of climate change.

A second limitation of previous models is that they often did not account for large-scale
climate forcing that affects multiple life stages and food webs simultaneously, and thus could
have compounding effects across developmental stages. Accounting for the correlation structure
of climate effects over the full life cycle is especially important for migratory species with
complex life histories^{22,23}. For example, freshwater processes affect salmon arrival timing to the
estuary, which in turn affects marine survival. We acknowledge that relationships between
survival and climate are often non-stationary²⁴, so we made the correlation structure of the
environmental drivers explicit, and in principle, flexible enough to incorporate future changes.

We used a stochastic age-structured life-cycle model^{25,26} with both density-dependent
and density-independent climate effects. Salmon survival was forced by environmental drivers
where future climate trends were based on ensemble projections from GCMs (Fig. 1). We used a
simulation framework to explore model assumptions and quantify different aspects of model
uncertainty, including the functional form of the model, covariate selection, and life stage-
specific sensitivity.

Fig. 1. Life cycle model diagram showing the life stages and climate drivers that
 influenced each stage. The freshwater stages include upstream migration and holding, spawning,
 rearing in tributaries, and migrating downstream. The migration passes through the Columbia
 River hydrosystem, bracketed by Lower Granite Dam (LGR) and Bonneville Dam (BON). The
 freshwater stages were influenced by stream flows (Flow) and temperatures (Temp). The marine
 stage, which has variable duration represented by multiple arrows, was influenced by ocean
 temperatures across the northeastern Pacific (SSTarc) and along the Washington coast (SSTwa).
 Other influences on ocean productivity are represented by Upwelling. Figure courtesy of Su Kim
 (NOAA Fisheries) and illustrations by Blane Bellerud (NOAA Fisheries).

We applied the model to eight populations within the Snake River spring/summer Chinook salmon Evolutionarily Significant Unit (ESU), which migrate downstream from their headwater habitat in central Idaho, past eight major hydroelectric projects in the Snake and Columbia rivers, to grow and mature in the northeastern Pacific Ocean (Fig. 2). They return to freshwater 1-4 years later and migrate upstream for a single spawning opportunity. We simulated population time series for eight exclusively wild populations within this ESU. The model reconstructs historical population dynamics closely for most populations (based on Komogorov-Smirnov diagnostics) and has been relatively stable when confronted with new data over the last 10 years, a period during which it has been used operationally by the National Oceanic and Atmospheric Administration (NOAA Fisheries). This updated version of the model is similar in general characteristics to Crozier et al.²⁵ but imposes climate influences in more life stages and has stronger validation based on data from over a million individually tagged fish (Supplementary Table S1).

Fig. 2. Map of the Columbia River Basin (top), showing the location of the Salmon River
Basin (in red), Bonneville Dam (BON) and Lower Granite Dam (LGR). Modeled populations in
the salmon river basin (lower map) are shown by green boxes in the lower map and populations
are named after their natal stream.

In our results, climate change in the ocean was particularly catastrophic. We compared
multiple alternative covariates in survival models, acknowledging that the best predictor might
change over time and thus track different temporal trajectories²⁴. Notably, sea surface

temperature (SST) is an important component of most relevant indices ²⁷, as it reflects complex
interactions between atmospheric forcing, wind strength, upwelling, and mixing of ocean layers,
all of which affect productivity throughout the California Current Ecosystem²⁸. Although the
response of these processes to greenhouse gas forcing will vary, SST will likely continue to be
an important indicator, and SST will increase with climate change ^{29,30}.

Results

Quasi-extinction thresholds surpassed in warming climate

We assessed the first year (if any) in which a population in a given simulation fell below
a quasi-extinction threshold of adult abundance (QET50). The QET50 is passed when the
running mean of spawners, measured at the spawning stream, drops below 50 individuals in any
4-year period ³¹. Although small populations drop below this threshold periodically even in a
stationary climate, larger populations do not. The proportion of all simulations in which a
population dropped below the quasi-extinction threshold (QET50) increased dramatically under a
warming climate compared with a detrended (stationary) climate (Fig. 3). Under the
representative concentration pathway (RCP 8.5) ensemble mean projection, even the largest
populations (Bear Valley Creek and Secesh River) fell below QET50 in over 75% of simulations
by 2060. In RCP 4.5, the same milestone was passed a decade later.

Fig. 3. The percent of simulations in which individual populations reached the quasi-extinction threshold (QET50) by a given year under scenarios assuming extensive interannual and decadal variability but no consistent trends across simulations (a stationary climate) vs. scenarios where trends were superimposed on this natural variability. Trends reflect the median GCM projection (solid lines) and the interquartile range of GCM projections for a given emissions scenario (shading).

Extinction rates increased under the climate change scenarios because population abundances diverged quickly in the climate scenarios compared with the baseline conditions represented by a stable climate (Fig. 4).

Fig. 4. Population abundance over time. Median (lines) and interquartile range (shading)

of population abundance for the ensemble mean GCM projection per RCP in relation to QET50

(dashed horizontal line). Note the different y-axes for small (top row) and large (bottom row)

populations.

Similar results with alternative model covariates

We conducted sensitivity analyses to characterize the how population outcomes
(extinction risk) varied depending on the covariates that were included in the model, and the
extent to which impacts in different life stages predicted population-level responses.

Alternative marine and freshwater covariate models had relatively little effect on overall
patterns (Fig. 5). Resulting from the high importance of SST in model comparisons, all three
models included combinations of basin-wide and coastal SST indices, which drives much of the
overall effect on survival²⁷. The freshwater models differed in whether fall (Models 1 and 3) or
summer flows (Model 2) limited smolt productivity. We considered both because the data were
consistent with both models (Supplementary Table S3). Populations fared slightly worse in the
summer-flow models because of projected decreases in summer precipitation, increases in
evaporation, and reduced groundwater storage; while fall precipitation (and hence fall flow) is
projected to either stay the same or increase (Fig. S2). The models that included upwelling (blue)
were more optimistic in more aggressive warming scenarios (GCM₇₅) because some GCMs
project a positive trend in spring upwelling later this century in the northern California Current
Ecosystem (Fig. S2). Nonetheless, negative effects from SST still drove most populations extinct
within the century (Fig. 5).

Fig. 5. Year that each population fell below QET50 for three models. Different models
are shown in different colors (Model 1 (red): covariates include SST and summer flow; Model 2
(yellow): covariates include SST, summer temperature and fall flow; Model 3 (blue): covariates
include SST, upwelling, summer temperature and fall flow). Extinction happens earlier in
scenarios with more warming (GCM₇₅) in models without upwelling (yellow and red) but later
when more intense upwelling ameliorates SST effects (blue). However, in all models extinction
happens earlier in the RCP 8.5 scenarios compared with a stationary climate. Note that larger
populations (bottom row) often never dropped below QET50 in the stationary climate, indicated
with ~.

**Marine life stage most vulnerable to warming**

The impact of climate change on freshwater life stages was not always negative, but
rather depended on population-specific and model-specific sensitivity. In the headwaters, the
model with fall flow produced a net benefit to parr to smolt survival, whereas the summer flow
model lowered survival (red and blue boxes vs. yellow in juvenile rearing stage in Fig. 6), but
produced no difference in effects at other life stages. Thus if climate change affected only the
early rearing period and no other stage, there could be a mix of responses across populations
depending on local limiting factors such as summer vs. fall flow, similar to the conclusions of
Crozier et al. ²⁵.

Fig. 6. A three by four factorial experimental design examining the relative loss in spawner abundance from 2020 to 2060 due to the effects climate change. Net trends in population size are shown using the ratio of late abundance (2060-2069) compared with early abundance (2020-2029) when RCP 8.5 ensemble mean climate trends are imposed during a single life stage only, with other life stages under a stationary climate. The horizontal line ($Y=1$) indicates no change in spawner abundance. Boxes show the interquartile range across simulations, while the whiskers extend to the most extreme values of individual simulations. Model 1 covariates include SST and summer flow; Model 2 covariates include SST, summer temperature and fall flow; Model 3 covariates include SST, upwelling, summer temperature and fall flow.

During the spawning migration, summer-run populations (i.e., populations that returned
to their spawning areas in summer, Secesh River and Valley Creek) were more affected by
temperature than spring-run populations, with net declines of up to -17% by the 2060s. However,
temperature effects on the juvenile, downstream migration reduced populations by about -18%
from the 2020s to the 2060s, on average, while climate change effects in the marine stage
reduced survival by -83% to -90% (Fig. 6).

It is important to note that the freshwater climate impacts in this study were conservative
in some respects. Specifically, we have assumed linear responses to environmental variables, but
physiological and ecological thresholds can create non-linear responses, by which future
temperatures or flows could have more severe effects. Furthermore, the primary covariate for
juvenile survival in two of our models was fall flow, which is the covariate that is least sensitive
to climate change³². Summer flows could also be limiting (Model 2), which would cause a more
negative response. More generally, the high elevation, mostly-wilderness habitat of these
populations is unusual for salmon in the region, and partially explains the relatively small effects
of climate change on their freshwater life stages. Other populations face more immediate impacts
on freshwater productivity^{14,33}.

Survival through the migration corridor declined for all juvenile migrants and adult
summer-run migrants due to rising temperatures. Still, the declines we found were relatively
small because of their early run timing compared with other salmon that migrate during peak
temperatures. In particular, endangered Snake River sockeye adults experience much higher
mortality from heat stress^{34,35}. In our analysis, we found relative resilience in freshwater stages

and the dominant driver towards extinction was the ~90% decline in survival due to rising SST
in the marine life stage. Therefore, closely monitoring ocean survival and directing research into
these populations potential response to novel conditions is clearly needed.

**Caveats**

There are two main caveats to these projections. First, the northeast Pacific might not
warm at the rate modeled, despite rising levels of CO₂ in the atmosphere. Over the past century,
internal variability in the climate system represented by variation in sea level pressure and
natural variability in ocean circulation has been a stronger determinant of coastal SST than
global mean temperature³⁶. How long this situation will continue is difficult to predict. Warming
might occur slower than modeled, which would reduce the rate of population declines.
Nonetheless, with the entire ocean warming at all depths³⁰, at some point this signal will
inevitably reach coastal waters.

The second possibility is that the northeast Pacific does warm, but some sort of
ecological surprise³⁷ will reverse the historical relationship between SST and salmon survival.
Ocean temperature does not affect salmon primarily through a physiological response, but rather
through a combination of bottom up and top down ecological processes that jointly regulate
salmon growth and survival³⁸⁻⁴⁰, which explains the non-stationarity of statistical correlations²⁴.
Warm conditions have been associated with poorer-quality prey and more warm-water predators,
likely generating the correlation we have observed. However, it is possible that novel
communities will arise with different responses to temperature, or that salmon will adapt to an
altered food web in a positive manner. We do see consumption rates increase and unexpected
species appear in the California Current Ecosystem when new conditions arise, such as during

the marine heatwave of 2013-2015⁴¹. For example, anchovy (*Engraulis mordax*), sardine
(*Sardinops sagax*) and hake (*Merluccius productus*) showed unusually early and northern
spawning behavior, which increased concentrations of larvae in the northern California Current
Ecosystem in the winter 2015 and 2016, a shift that could benefit salmon⁴².

Nonetheless, the correlation strength with SST has been increasing rather than decreasing
252⁴³, and in fact salmon fared poorly⁴⁴ during the recent marine heatwave, with a decadal-low
number of adult Chinook returning in many ESUs⁴⁵ and the closure of multiple fisheries in
2020. Thus, although the various processes that historically generated the PDO²⁰ may interact
differently in the future, it seems likely that SST will continue to be a negative indicator of
salmon survival. Further exploration using our model with a changing correlation structure over
time could clarify possible trajectories and when they could be detected.

Other ecological surprises should also be considered, such as increases in competitors
such as jellyfish⁴⁰ and Humboldt or market squid⁴⁶, which could reduce salmon survival in an
altered ocean. Predators, such as seabirds that currently concentrate on alternative prey, change
behavior when their preferred prey dwindle or alter their distribution, which can increase or
decrease predation on salmon⁴⁷. We recommend closely monitoring trophic interactions and
salmon growth rates to detect such a possibility.

Finally, our model is conservative in that we have not accounted for any negative effects
of ocean acidification. Declines of sensitive species such as crabs and calciferous zooplankton
could have a negative effect on salmon, especially salmon populations that prey extensively on
sensitive species⁴⁸. We have assumed that ocean-stage salmon are relatively insensitive to pH,
but if there are effects, they will likely be negative^{49,50}.

Discussion

Our results indicate that rising SST as one symptom of a changing ocean puts all of these
populations at high risk of extinction. Small populations have minimal buffer against declining
marine survival rates, and are at immediate risk (Fig. 1). The threat to larger populations causes
even greater concern because they are the remaining salmon strongholds, which provide genetic
and demographic resilience for the ESU as a whole⁵¹. Moreover, although we have focused on
the details for certain populations, strong synchrony across all populations within the ESU⁵² and
more broadly^{53,54} suggests these responses could represent a widespread phenomenon. Salmon
populations show coherence at many spatial scales, and synchrony has increased over time in
both Pacific and Atlantic populations and climate indicators^{3,55,56}. To the extent that negative
responses to SST continue as a shared theme in these patterns^{57,58}, climate change overlaid on
other anthropogenic forces⁵⁹ could drive salmon declines at a large scale.

There is no easy way to mitigate for increasing SST that produce declines in marine
survival of this magnitude. Ecosystem-based fisheries management attempts to consider the
complex web of drivers with both direct and indirect links to anthropogenic actions, and track
indicators in all sectors^{45,60}. Conceptual models of factors that affect salmon marine survival
show a large number of interacting processes^{61,62}. Importantly, solutions involving management
actions could occur in either marine or freshwater realms.

In the marine realm, human activities affect salmon survival through targeted fishing on
salmon, their prey (sardine, anchovy, krill, juvenile rockfish, juvenile crab), predators (e.g.,
marine mammals) and competitors (e.g., hake, Pacific halibut, arrowtooth flounder). Changes in
fisheries can have positive or negative effects on salmon. Curtailed salmon harvest since the

1960s reduced one major source of direct mortality. Reduced harvest on top predators, however,
led to rebounding marine mammal populations that now consume large amounts of salmon^{39,63}.
Competition with hatchery fish has complex interactions with climate effects on wild salmon⁵⁹,
and is an area of active research. Sea level rise combined with coastal development threatens
complete loss of intertidal marshes in California and Oregon, and the majority of estuary habitat
in Washington⁶⁴, which will affect some salmon and their prey.

Other indirect anthropogenic effects on marine habitats are not well quantified, and
warrant more research. Increased awareness of the importance of forage fish for the entire food
chain⁶⁵ led to a ban on the development of fisheries to exploit forage fish, demonstrating a
proactive approach that should support salmon. But in sum, we lack key information on the full
mechanistic basis of salmon marine survival, which limits the strength of end-to-end models for
guiding management⁶⁶.

Efforts to mitigate carryover effects from freshwater that could affect marine survival in
these populations have primarily focused on dams. Survival through Columbia and Snake River
dams generally now meets recovery targets (>96%)⁶⁷, and cumulative mortality over 500 km of
in-river migrating fish (~50%) is similar to that estimated for unregulated rivers of similar length
(i.e., Fraser River⁶⁸). However, slow travel time through reservoirs combined with temperatures
that have been elevated by dams⁶⁹ can potentially result in lower marine survival⁷⁰. Mitigation
efforts to increase smolt body size and advance migration timing could increase marine survival
310^{71,72}. Restoration efforts in freshwater habitat, such as restoring floodplains, riparian planting to
311 reduce stream temperature, reconnecting side-channel habitat⁷³ or adding nutrients to juvenile
salmon rearing areas could also enhance/restore freshwater salmon production.

Our models estimated strong density dependence at the parr to smolt stage, although it is
not clear whether summer or winter habitat is constrained. The headwaters have minimum
anthropogenic impacts, but the mainstem Salmon River has experienced structural simplification
and loss of wood, which could limit both rearing and overwintering capacity. Our results suggest
that smolt carrying capacities are currently limited by flow rather than temperature. Higher flows
may create more habitat, improve connectivity, or decrease contact with predators. The predator
community has also been affected by human impacts, from introduced sport fish (smallmouth
bass, *Micropterus dolomieu* and brook trout *Salvelinus fontinalis*) to creation of reservoir habitat
more favorable for invasive fish (e.g., American shad, *Alosa sapidissima*).

Throughout salmon watersheds, improving and expanding access to rearing habitat
should increase smolt abundance and body condition resulting in improved salmon viability ⁷⁴.
Intrinsic habitat potential is negatively correlated with current levels of disturbance, so restoring
habitat could yield substantial benefits. Specifically, habitat at lower elevation that was
historically highly productive has been preferentially lost. Improving individual fish growth by
reducing contaminant loads ⁷⁵, increasing floodplain habitat ⁷⁴ and habitat complexity in general
could boost population productivity ⁷⁶.

Prospects for saving this iconic keystone species in the conterminous U.S. are
diminishing. Resilience to climate change depends on genetic and ecological diversity to adapt to
environmental change. Many options have been considered over decades of dedication to salmon
recovery, and improvements have been made. However, the urgency is greater than ever to
identify successful solutions at a large scale and implement known methods for improving

survival. Management actions that open new habitat, improve productivity within existing
habitat, or reduce mortality through direct or indirect effects in the ocean are desperately needed.

Methods

Data informing survival estimates

Spawner age and abundance estimates were compiled through large-scale collaborative
 efforts between states, tribes, and coordinating bodies⁷⁷⁻⁷⁹. Stage-specific survival estimates
 were obtained from multiple sources^{27,80-84}, originating from tagging and detection records
 downloaded from PTAGIS.org (Supplementary Table S1). We only used detection records for
 fish identified as wild from known population sources. Environmental covariates used in the
 model include air temperature, stream flow and temperature, SST, and coastal upwelling
 (Supplementary Table S2).

Life cycle model structure

We employed a stochastic, age-structured model modified from^{25,26,85}. The model as
 depicted in Fig. 7 has five annual time steps, based on the five-year generation time of Snake
 River spring/summer Chinook salmon, which correspond approximately to life-stage transitions.
 The first time step (“spawner to parr” stage) spans fall spawning(months), incubation and early
 parr rearing. Survival through the second time step (S_2) includes both tributary rearing ($S_{tributary}$)
 from summer (July or August) to the following spring (months)when they pass Lower Granite
 Dam, and migration ($S_{mainstem}$) through the Snake and Columbia River hydrosystem. The third
 time step includes ocean entry and the first winter and spring in the ocean. Some Chinook
 salmon return to spawn in their third year (jacks), but most females and “adult” males stay in the
 ocean for one or two more years, during which ocean survival is represented as S_o . The number
 of fish in the ocean each year is a latent variable, fit by detections of survivors when they re-
 enter freshwater (“smolt to adult return”, S_{sar}). Upstream migration survival through the

hydrosystem ($S_{upstream}$) and from Lower Granite Dam to spawning ($S_{prespawn}$) are captured in the
 fourth or fifth time step. Older females tend to lay more eggs, which is reflected in the fecundity
 parameter, F_5 . To calculate “effective spawners”, we combined different age classes that
 returned to spawn in the same year as the weighted sum of 3-y-old (weight=0), 4-y-old
 (weight=1), and 5-y-old (weight= F_5) fish, which is estimated by an expansion of the number of
 redds (nests) counted during spawning surveys ³¹.

Figure 7. Diagram of life cycle model. Environmental covariate survival models that
 were fit directly to PIT-tagged data are shown in blue ($S_{tributary}$, $S_{mainstem}$, S_{SAR} , $S_{upstream}$). Life stages
 and fitted transition parameters are in black, with associated equations for reference in the
 supplementary material.

**Model fitting**

We fit the life cycle model in two steps. We first fit individual life stage relationships
 with covariates using a variety of methods in different life stages, generating a posterior
 distribution for stage-specific parameter estimates. In addition to these parameters, the life cycle
 model introduces some additional parameters that could not be directly fit to data. We therefore
 conducted a second step to calibrate the life cycle model using a modified Approximate Bayesian
 Computing approach^{86,87}. In the calibration step, we simulated population time series under
 recent climatic conditions using the life cycle model, and compared the resulting spawner and
 smolt time series to those observed, and ranked parameter sets by the deviance from a
 Kolmogorov-Smirnov test. We selected the top 0.2% of 500,000 sets of parameter combinations.
 All of the resulting parameter sets met the criterion of producing a spawner distribution that was
 not statistically different from the observation dataset (p-value>0.05 from the Kolmogorov-
 Smirnov test). This process maintains the appropriate correlation structure within each parameter
 set.

**Spawner to parr (S_I) and parr to smolt ($S_{tributary}$) stages:** We fit adult recruits per
 spawner for eight populations in a hierarchical Bayesian framework using multiple likelihood
 equations that reflected stages that could be compared directly with data. Briefly, we fit a 2-stage
 Gompertz function^{88, see Supplementary Methods} to solve the two stages simultaneously, combined with
 independently-estimated survivals for later stages. Individual population coefficients
 (productivity and capacity parameters for both stages as well as coefficients for temperature and
 flow) were assumed to be random samples from an underlying normal distribution⁸⁹. To
 determine the best environmental covariates, we compared the estimated predictive error of
 alternative models in a leave-one-out cross-validation method for Bayesian models using the

LOO package⁹⁰. Because of correlations among the climate variables tested, multiple models
had similar support from the data. Our primary objective was to identify divergent potential
responses to climate change. Therefore, we selected two models with covariates that show
different trajectories with climate change (Fig. S2). Model 1 included summer air temperature
and fall stream flow, while model 2 included summer stream flow only.

**Juvenile survival through the Columbia River hydrosytem ($S_{mainstem}$):** We estimated
juvenile survival from Lower Granite Dam to Bonneville Dam and arrival day at Bonneville
Dam using the COMPASS model⁹¹. In the COMPASS model, each dam and riverine reach has
equations that use hourly flow, temperature, and spill to predict fish survival, migration rate, and
the proportion of fish that use the spillway, turbine or bypass passage route at dams. The
COMPASS model also tracks the proportion of fish that were loaded into barges to bypass
migration through the hydropower system.

**Smolt to adult return (S_{sar}):** We used a mixed-effects logistic regression model²⁷ to
determine the effect of the date of ocean entry and environmental covariates on the probability
that an individual fish would return as an adult to Bonneville Dam. The model includes random
effects of day and a day by year interaction, which follow an auto-regressive process. Using
Akaike Information Criterion, we selected variables with high importance based on model
weights, which included a large-scale measure of SST (SSTarc) and a more local, coastal
measure of SST (SSTwa), as well as spring upwelling. We applied separate models for fish that
had migrated through the mainstem in the river and for fish that had been transported
downstream (Supplementary Table S4).

**Adult upstream survival ($S_{upstream}$):** For the adult upstream survival model, we used
 generalized additive mixed models (GAMMs) to evaluate the effects of both anthropogenic and
 environmental covariates on spring/summer Chinook salmon survival⁹². To run the model in
 simulation mode, all the non-environmental covariates (fisheries catch, the proportion of fish that
 had been transported in barges as juveniles) had similar distributions to the baseline period 2004-
 2016. We held survival from the hydrosystem to spawning ($S_{prespawn}$) constant due to the lack of
 appropriate data for most populations with which to fit a relationship.

**Calibration step:** The model includes a set of maturation parameters that could not be
 estimated directly from the data, so we treated them as tuning parameters for the life cycle model
 as a whole. These parameters partition total smolt-to-adult survival (SAR) to age-specific
 survivals (S_3) for the first year, then S_0 for the 2nd and 3rd year), combined with a propensity to
 return at a given age (jacks: b_3 , 4y olds: b_4 , 5 y olds: $1-b_4$) as follows:

$$431 \quad S_{SAR} = \frac{b_3 * N_3 + b_4 * N_4 + N_5}{N_2} \quad [S11]$$

Where $N_3 = S_3 * N_2$

$$N_4 = (1 - b_3) * N_3 * S_0$$

$$N_5 = (1 - b_4) * N_4 * S_0$$

Fish that stay in the ocean longer have additional mortality (S_0). There is still an
 advantage to spawning as an older fish because of higher fecundity (F_5). In the model, the
 effective number of spawners reflects the age distribution of female spawners, which return as
 either 4 or 5 y olds, and the 5 y olds have the fecundity advantage. A very small percentage of
 fish return as 6 y olds; these fish were added to 5 y olds.

We fit the tuning parameters (S_0 , b_3 , b_4 , and F_5) using a modified Approximate Bayesian
Computing approach^{86,87}. We applied this method by first generating a prior distribution for
each parameter. The priors for parameters that range between 0 and 1 (S_0 , b_3 , b_4) had a beta
distribution centered on the mean and including the full range used by Zabel et al.^{26, Table 1}. F_5
had a normal distribution with mean from Kareiva et al.⁸⁵ and $sd=0.1$. These values were
randomly combined with the freshwater parameter sets from the posterior distribution from the
Bayesian analysis to create 500,000 combinations of all 10 parameters.

We ran the life cycle model using each of these parameter sets in a different simulation
(500,000 iterations per population). The life cycle model was forced by historical meteorological
conditions from 2000 to 2015 (Table S2). For juvenile mainstem survival, we used COMPASS
reconstructions of juvenile migration that incorporated actual river management. Note that
historical river management was extremely variable over this time period and not comparable to
the conditions we projected in the climate simulations (in particular, transportation rates were
higher, spill was lower, and dam passage survival was lower than in the action proposed in the
Environmental Impact Statement ACOE⁹³). We ran the SAR model in retrospective mode,
which uses the fitted estimates of historical random effects and observation error. We simulated
other life stages in the calibration as we would in future projections.

The parameter values from top 0.2% of these parameter sets compared with their prior
distributions are shown in Fig. S1. The resulting models produced Kolmogorov-Smirnov p-values
from 0.16 to 0.76, which all rejected the null hypothesis that the modeled and observed time
series came from different distributions ($p<0.05$).

**Climate scenarios**462 **Stationary climate simulations**

When creating future climate scenarios, our aim was to maintain the statistical properties
 of the environmental data driving survival in the various life stages. Large-scale oceanic and
 atmospheric drivers affect the marine environment and the freshwater environment
 simultaneously. To account for this, we estimated an unstructured covariance matrix for all the
 freshwater and marine environmental covariates (Supplementary Table S5) using the TMB
 libraries for R⁹⁴. We used a multivariate state space model

$$469 \quad \mathbf{x}_t \sim \text{MVN}(\rho^x \mathbf{x}_{t-1}, \mathbf{Q}) \quad [3]$$

$$470 \quad \mathbf{y}_t \sim \text{N}(\mathbf{x}_t, 0.001) \quad [4]$$

where \mathbf{x}_t is a vector of the environmental processes at time t , ρ^x is the correlation between the
 vectors from successive time steps, and \mathbf{Q} is an unstructured covariance matrix. For n covariates,
 the number of estimated correlation coefficients in the unstructured covariance matrix is equal to
 $0.5 * n * (n - 1)$. The observation error for the observation model was fixed to 0.001, implying
 the covariate data were measured essentially without error.

**Climate change scenarios**

Our objective was to explore specifically how the relationships among various climate
 drivers interacted with population dynamics to shape population trajectories. To do this, we first
 simulated 1,000 time series of 75 years of stationary environmental conditions that followed the
 observed covariance relationships in the historical record. Second, we added offsets to these
 stationary simulations according to GCM projections. Each offset (trend) consisted of a single

time series added to all 1,000 stationary simulations. We then input the resulting combinations of
stationary-plus-offset environmental conditions for each climate scenario into the life cycle
model as forcing factors.

For each model/population/climate scenario, we ran 1,000 iterations. Climate trends
were created from the median (GCM₅₀) and interquartile range across 10 to 80 GCM projections,
depending on the covariate (Supplementary Table S6). We tracked the geometric mean of
population abundance in 10-year intervals. We also assessed the first year (if any) in which a
population in a given simulation fell below a quasi-extinction threshold of adult abundance
(QET50). The QET50 is passed when the running mean of spawners, measured at the spawning
stream, drops below 50 individuals in any 4-year period³¹.

We explored projected climate trends from two carbon emissions scenarios,
representation concentration pathway (RCP) 4.5 and 8.5^{95,96}. We used time series output from
GCMs directly for the marine variables, and output that had been statistically downscaled then
processed through hydrological and stream temperature models for the freshwater variables
496^{32,97,98}. For each variable, 10 to 80 time series per emissions scenario were available
(Supplementary Table S6). We smoothed each of these individual time series using a 20-y
running mean to reduce interannual variation that was already accounted for by the TMB model.

From these resulting time series, we calculated the 25th, 50th, and 75th quantiles each year
to generate three trends for each covariate in each climate scenario. These trends represent the
spread across climate models of low, medium, and high rates of change in each covariate. Thus
in summary, we added each climate trend to the raw simulation of a stationary climate produced

by the covariance model. The new time series retained the autocorrelation, variance and
covariance of the historical environment with forcing from greenhouse gas emissions added.

**Life cycle model simulations**

Running the model in simulation mode was relatively straightforward for the stages that
were fit on an annual time step (S_I and $S_{tributary}$) because they only required an annual input of
environmental conditions. However, submodels for upstream and downstream survival required
daily environmental inputs. We therefore ran a separate step to reproduce numerous
representations of daily time series, which were then grouped by their annual spring mean
temperatures and flows. For each annual time step, we sampled randomly from the appropriate
disaggregated submodel output (Supplementary Methods S2).

We ran the life cycle model from 2015 to 2089, applying climate trends to the stationary
climate for each environmental covariate on an annual time step. We repeated 1,000 iterations
515 per population per model per climate scenario. Our first results stem from a single combination
of freshwater and marine covariates (Model 1: summer temperature + fall flow for freshwater,
SSTarc in winter for transported fish, and SSTarc in winter + SSTwa in summer for in-river fish)
to show the full time series of population response to the climate scenarios (Fig. 1-2).

**Sensitivity analyses**

In our first sensitivity analysis, we compared population outcomes from Model 1 with a
different freshwater covariate model (Model 2: summer flow + Model 1 marine covariates).
Finally, we exchanged the marine covariates in Model 3: summer temperature + fall flow for
freshwater, SSTwa in summer for transported fish, and SSTarc in spring + spring upwelling for

in-river fish (Fig 3). Thus, we explored all combinations of the top two models for freshwater
and marine stages, respectively.

In our second sensitivity analysis, we applied the climate trends for the ensemble mean of
RCP 8.5 to one life stage while the other life stages experienced a stationary climate (Fig. 6). We
cycled through parr to smolt, downstream migration, smolt to adult return, and upstream
migration. In each case, we reported the extent of population decline as the ratio of geometric
mean population size in 2080-2089 divided by mean abundance in 2020-2029. We ran 1,000
simulations per life stage, and calculated the mean change in abundance for each population.

**Acknowledgements:**

This work would not have been possible without the enormous effort that went into PIT-
tagging fish, designing, building and maintaining detection facilities, conducting redd counts,
and managing the various databases. State (Idaho, Oregon, and Washington), federal (U.S. Fish
and Wildlife and NOAA Fisheries), and tribal entities (especially Nez Perce) PIT-tagged most of
the fish used in this study. IDFG and Nez Perce conducted most of the redd counts, and Mari
Brick assisted in managing the spawner database. We thank Steve Achord, Gordon Axel, and
Jesse Lamb, from the Wild Fish Monitoring Program at NOAA Fisheries for their focus on
tagging wild fish in natal river reaches. Susannah Iltus (Columbia Basin Research) assembled
PIT tag data according to juvenile transportation history. We also thank Jeff Jorgenson, Jennifer
Gosselin, and Eric Ward for reviewing drafts of the manuscript.

**Author contributions**

The primary author contributed to the conception, design, analysis and interpretation of
data and writing the paper. BB and RZ contributed to design and interpretation of the work and
manuscript revisions; BC contributed to analysis, developing new methods and writing. DW led
data acquisition from PTAGIS, design and analysis.

**Competing interests**

No authors have competing interests in these results. NOAA Fisheries funds the salaries
of all authors but has no influence on the results.

**List of Supplementary Methods and Materials:**

Supplementary Methods S1. Submodels for individual life stages

Table S1. Biological data sources

Table S2. Environmental data sources

Table S3. Model comparison of spawner to smolt productivity models.

Table S4. SAR models selected

Figure S1. Posterior distributions of parameter estimates used in life cycle model

simulations

Supplementary Methods S2. Aggregation of upstream and downstream submodels to an annual
time step

Supplementary Methods S3. Climate trends

Table S5. Correlation of environmental covariates

Table S6. Climate trend sources

Figure S2. Trends from GCM projections

Literature cited

- Nicola, G. G., Elvira, B., Jonsson, B., Ayllon, D. & Almodovar, A. Local and global climatic drivers
of Atlantic salmon decline in southern Europe. *Fisheries Research* **198**, 78-85,
doi:10.1016/j.fishres.2017.10.012 (2018).
- Peterman, R. M. & Dorner, B. A widespread decrease in productivity of Sockeye Salmon
(*Oncorhynchus nerka*) populations in western North America. *Canadian Journal of Fisheries and*
*Aquatic Sciences*, 1255–1260 (2012).
- Chaput, G. Overview of the status of Atlantic salmon (*Salmo salar*) in the North Atlantic and
trends in marine mortality. *ICES Journal of Marine Science* **69**, 1538-1548,
doi:10.1093/icesjms/fss013 (2012).
- Galbreath, P. F., Bisbee, M. A., Dompier, D. W., Kamphaus, C. M. & Newsome, T. H. Extirpation
and tribal reintroduction of Coho salmon to the interior Columbia River basin *Fisheries* **39**, 77-
87, doi:10.1080/03632415.2013.874526 (2014).
- Wasser, S. K. *et al.* Population growth is limited by nutritional impacts on pregnancy success in
endangered Southern Resident killer whales (*Orcinus orca*). *Plos One* **12**,
doi:10.1371/journal.pone.0179824 (2017).
- NMFS, National Marine Fisheries Service. West Coast salmon & steelhead listings. NOAA
Fisheries West Coast Region website available at
www.westcoast.fisheries.noaa.gov/protected_species/salmon_steelhead/salmon_and_steelhead_listings/salmon_and_steelhead_listings.html (December 2016). (2014).
- NRC, Committee on Protection and Management of Pacific Northwest Anadromous Salmonids.
*Upstream : salmon and society in the Pacific Northwest*. Vol. Board on Environmental Studies
and Toxicology. Commission on Life Sciences (National Academies Press, 1996).
- Katz, J., Moyle, P. B., Quinones, R. M., Israel, J. & Purdy, S. Impending extinction of salmon,
steelhead, and trout (Salmonidae) in California. *Environmental Biology of Fishes* **96**, 1169-1186,
doi:10.1007/s10641-012-9974-8 (2013).
- Kovach, R. P. *et al.* Genetic diversity is related to climatic variation and vulnerability in
threatened bull trout. *Global Change Biol.* **21**, 2510-2524, doi:10.1111/gcb.12850 (2015).
- Simon, D. J. (2016).
- Link, J. S., Griffis, R. & Busch, S. (U.S. Department of Commerce, NOAA Technical
Memorandum NMFS-F/SPO-155, Silver Spring, MD, 2015).
- Crozier, L. G. *et al.* Climate vulnerability assessment for Pacific salmon and steelhead in the
California Current Large Marine Ecosystem. *PLOS ONE* **14**, e0217711,
doi:10.1371/journal.pone.0217711 (2019).
- Hare, J. A. *et al.* A vulnerability assessment of fish and invertebrates to climate change on the
northeast U.S. continental shelf. *PLOS ONE* **11**, e0146756, doi:10.1371/journal.pone.0146756
(2016).
- Honea, J. M., McClure, M. M., Jorgensen, J. C. & Scheuerell, M. D. Assessing freshwater life-stage
vulnerability of an endangered Chinook salmon population to climate change influences on
stream habitat. *Climate Research* **71**, 127-137, doi:10.3354/cr01434 (2017).
- Battin, J. *et al.* Projected impacts of climate change on salmon habitat restoration. *Proceedings*
*of the National Academy of Sciences of the United States of America* **104**, 6720-6725,
doi:10.1073/pnas.0701685104 (2007).

- Thompson, L. C. *et al.* Water Management Adaptations to Prevent Loss of Spring-Run Chinook
Salmon in California under Climate Change. *Journal of Water Resources Planning and*
*Management* **138**, 465-478, doi:doi:10.1061/(ASCE)WR.1943-5452.0000194 (2012).
- Cheung, W. W. L., Brodeur, R. D., Okey, T. A. & Pauly, D. Projecting future changes in
distributions of pelagic fish species of Northeast Pacific shelf seas. *Progress in Oceanography*
**130**, 19-31, doi:10.1016/j.pocean.2014.09.003 (2015).
- Morley, J. W. *et al.* Projecting shifts in thermal habitat for 686 species on the North American
continental shelf. *PLOS ONE* **13**, e0196127, doi:10.1371/journal.pone.0196127 (2018).
- Burke, B. J. *et al.* Multivariate models of adult Pacific salmon returns. *PLoS One* **8**, e54134,
doi:10.1371/journal.pone.0054134 (2013).
- Newman, M. *et al.* The Pacific decadal oscillation, revisited. *Journal of Climate* **29**, 4399-4427,
doi:10.1175/jcli-d-15-0508.1 (2016).
- Li, S. *et al.* The Pacific Decadal Oscillation less predictable under greenhouse warming. *Nature*
*Climate Change* **10**, 30-34, doi:10.1038/s41558-019-0663-x (2020).
- Crozier, L. G. *et al.* Potential responses to climate change in organisms with complex life
histories: evolution and plasticity in Pacific salmon. *Evolutionary Applications* **1**, 252-270 (2008).
- Healey, M. The cumulative impacts of climate change on Fraser River sockeye salmon
(*Oncorhynchus nerka*) and implications for management. *Canadian Journal of Fisheries and*
*Aquatic Sciences* **68**, 718-737, doi:10.1139/f11-010 (2011).
- Litzow, M. A. *et al.* Non-stationary climate-salmon relationships in the Gulf of Alaska.
*Proceedings of the Royal Society B-Biological Sciences* **285**, 9, doi:10.1098/rspb.2018.1855
(2018).
- Crozier, L. G., Zabel, R. W. & Hamlett, A. F. Predicting differential effects of climate change at the
population level with life-cycle models of spring Chinook salmon. *Global Change Biology* **14**, 236-
249, doi:10.1111/j.1365-2486.2007.01497.x (2008).
- Zabel, R. W., Scheuerell, M. D., McClure, M. M. & Williams, J. G. The interplay between climate
variability and density dependence in the population viability of Chinook salmon. *Conservation*
*Biology* **20**, 190-200 (2006).
- Chasco, B., Burke, B. J. & Crozier, L. G. Differential impacts of freshwater and marine covariates
on wild and hatchery Chinook salmon marine survival. *Plos One* (in review).
- Doney, S. C. *et al.* in *Annual Review of Marine Science, Vol 4* Vol. 4 (eds C. A. Carlson & S. J.
Giovannoni) 11-37 (2012).
- Alexander, M. A. *et al.* Projected sea surface temperatures over the 21st century: Changes in the
mean, variability and extremes for large marine ecosystem regions of Northern Oceans.
*Elementa-Science of the Anthropocene* **6**, doi:10.1525/elementa.191 (2018).
- IPCC. (ed D.C. Roberts H.-O. Pörtner, V. Masson-Delmotte, P. Zhai, M. Tignor, E. Poloczanska,
650 K. Mintenbeck, A. Alegría, M. Nicolai, A. Okem, J. Petzold, B. Rama, N.M. Weyer) (2019).
- ICTRT & Zabel, R. W. Required survival rate changes to meet Technical Recovery Team
abundance and productivity viability criteria for interior Columbia River basin salmon and
steelhead populations., Available at:
http://www.nwfsc.noaa.gov/trt/col_docs/ictrt_gaps_report_nov_2007_final.pdf (NWFSC,
Seattle, Washington, 2007).
- Chegwiddden, O. S. *et al.* How do modeling decisions affect the spread among hydrologic climate
change projections? Exploring a large ensemble of simulations across a diversity of
hydroclimates. *Earth's Future* <https://doi.org/10.1029/2018EF001047>, 7,
doi:10.1029/2018ef001047 (2019).

- Justice, C., White, S. M., McCullough, D. A., Graves, D. S. & Blanchard, M. R. Can stream and
riparian restoration offset climate change impacts to salmon populations? *J Environ Manage*
**188**, 212-227, doi:10.1016/j.jenvman.2016.12.005 (2017).
- Isaak, D. J. *et al.* Global warming of salmon and trout rivers in the northwestern U.S.: Road to
ruin or path through purgatory? *Transactions of the American Fisheries Society* **147**, 566-587,
doi:10.1002/tafs.10059 (2018).
- Crozier, L. G. *et al.* Migration survival of adult Snake River sockeye salmon through the Columbia
River Basin. *Research Report for U.S. Army Corps of Engineers, Walla Walla District. Available at*
https://www.nwfsc.noaa.gov/contact/display_staffprofilepubs.cfm?staffid=1471 (2018).
- Johnstone, J. A. & Mantua, N. J. Atmospheric controls on northeast Pacific temperature
variability and change, 1900-2012. *Proceedings of the National Academy of Sciences of the*
*United States of America* **111**, 14360-14365, doi:10.1073/pnas.1318371111 (2014).
- Lindenmayer, D. B., Likens, G. E., Krebs, C. J. & Hobbs, R. J. Improved probability of detection of
ecological “surprises”. **107**, 21957-21962, doi:10.1073/pnas.1015696107 %J Proceedings of the
National Academy of Sciences (2010).
- Ottersen, G., Kim, S., Huse, G., Polovina, J. J. & Stenseth, N. C. Major pathways by which climate
may force marine fish populations. *Journal of Marine Systems* **79**, 343-360,
doi:10.1016/j.jmarsys.2008.12.013 (2010).
- Chasco, B. E. *et al.* Competing tradeoffs between increasing marine mammal predation and
fisheries harvest of Chinook salmon. *Sci Rep* **7**, 15439, doi:10.1038/s41598-017-14984-8 (2017).
- Ruzicka, J. J., Daly, E. A. & Brodeur, R. D. Evidence that summer jellyfish blooms impact Pacific
Northwest salmon production. *Ecosphere* **7**, doi:10.1002/ecs2.1324 (2016).
- Morgan, C. A., Beckman, B. R., Weitkamp, L. A. & Fresh, K. L. Recent Ecosystem Disturbance in
the Northern California Current. *Fisheries* **44**, 465-474, doi:10.1002/fsh.10273 (2019).
- Auth, T. D., Daly, E. A., Brodeur, R. D. & Fisher, J. L. Phenological and distributional shifts in
ichthyoplankton associated with recent warming in the northeast Pacific Ocean. *Global Change*
*Biology* **24**, 259-272, doi:10.1111/gcb.13872 (2018).
- Ciannelli, Cunningham, Johnson & Puerta. Nonstationary effects of ocean temperature on Pacific
salmon productivity. *Can. J. Fish. Aquat. Sci.* **76**, 1923-1928, doi:10.1139/cjfas-2019-0120 (2019).
- Daly, E. A., Brodeur, R. D. & Auth, T. D. Anomalous ocean conditions in 2015: impacts on spring
Chinook salmon and their prey field. *Marine Ecology Progress Series* **566**, 169-182,
doi:10.3354/meps12021 (2017).
- Harvey, C. J. *et al.* Ecosystem Status Report of the California Current for 2019: A Summary of
Ecosystem Indicators Compiled by the California Current Integrated Ecosystem Assessment
Team (CCIEA). NOAA Institutional Repository: <https://doi.org/10.25923/mvhf-yk36>,
doi:<https://doi.org/10.25923/p0ed-ke21> (2019).
- Zeidberg, L. D. & Robison, B. H. Invasive range expansion by the Humboldt squid, *Dosidicus*
*gigas*, in the eastern North Pacific. **104**, 12948-12950, doi:10.1073/pnas.0702043104 %J
Proceedings of the National Academy of Sciences (2007).
- Wells, B. K. *et al.* Environmental conditions and prey-switching by a seabird predator impact
juvenile salmon survival. *Journal of Marine Systems* **174**, 54-63,
doi:10.1016/j.jmarsys.2017.05.008 (2017).
- Marshall, K. N. *et al.* Risks of ocean acidification in the California Current food web and fisheries:
Ecosystem model projections. *Global Change Biol.* **23**, 1525-1539 (2017).
- Ou, M. *et al.* Responses of pink salmon to CO₂-induced aquatic acidification. *Nature Clim.*
*Change* **5**, 950-955, doi:10.1038/nclimate2694

- <http://www.nature.com/nclimate/journal/v5/n10/abs/nclimate2694.html#supplementary-information>
 (2015).
- Williams, C. R. *et al.* Elevated CO₂ impairs olfactory-mediated neural and behavioral responses
 and gene expression in ocean-phase coho salmon (*Oncorhynchus kisutch*). **25**, 963-977,
 doi:10.1111/gcb.14532 (2019).
- McElhany, P., Ruckelshaus, M. H., Ford, M. J., Wainwright, T. C. & Bjorkstedt, E. P. Viable
 salmonid populations and the recovery of Evolutionarily Significant Units. Report No. Technical
 Memorandum NMFS-NWFSC 42, 156 (National Marine Fisheries Service, Northwest Fisheries
 Science Center, Seattle, WA, 2000).
- Jorgensen, J. C., Ward, E. J., Scheuerell, M. D. & Zabel, R. W. Assessing spatial covariance among
 time series of abundance. *Ecology and Evolution* **6**, 2472-2485, doi:10.1002/ece3.2031 (2016).
- Ohlberger, J., Scheuerell, M. D. & Schindler, D. E. Population coherence and environmental
 impacts across spatial scales: a case study of Chinook salmon. *Ecosphere* **7**,
 doi:10.1002/ecs2.1333 (2016).
- Zimmerman, M. S., Irvine, J. R., O'Neill, M., Anderson, J. H., Greene, C. M., Weinheimer, J., ... &
 Rawson, K. Spatial and temporal patterns in smolt survival of wild and hatchery coho salmon in
 the Salish Sea. *Marine and Coastal Fisheries* **7**, 116-134 (2015).
- Dorner, B., Catalano, M. J. & Peterman, R. M. Spatial and temporal patterns of covariation in
 productivity of Chinook salmon populations of the northeastern Pacific Ocean. *Canadian Journal*
 *of Fisheries and Aquatic Sciences* **75**, 1082-1095, doi:10.1139/cjfas-2017-0197 (2018).
- Black, B. A. *et al.* Rising synchrony controls western North American ecosystems. *Global change*
 *biology* **24**, 2305-2314 (2018).
- Mills, K. E., Pershing, A. J., Sheehan, T. F. & Mountain, D. Climate and ecosystem linkages explain
 widespread declines in North American Atlantic salmon populations. *Glob Chang Biol* **19**, 3046-
 3061, doi:10.1111/gcb.12298 (2013).
- Litzow, M. A., Ciannelli, L., Cunningham, C. J., Johnson, B. & Puerta, P. Nonstationary effects of
 ocean temperature on Pacific salmon productivity. *Can. J. Fish. Aquat. Sci.* **76**, 1923-1928,
 doi:10.1139/cjfas-2019-0120 (2019).
- Cline, T. J., Ohlberger, J. & Schindler, D. E. Effects of warming climate and competition in the
 ocean for life-histories of Pacific salmon. *Nat Ecol Evol* **3**, 935-942, doi:10.1038/s41559-019-
 0901-7 (2019).
- Andrews, K. S. *et al.* The legacy of a crowded ocean: indicators, status, and trends of
 anthropogenic pressures in the California Current ecosystem. *Environ. Conserv.* **42**, 139-151,
 doi:10.1017/S0376892914000277 (2015).
- Harvey, C. J., Reum, J. C. P., Poe, M. R., Williams, G. D. & Kim, S. J. Using Conceptual Models and
 Qualitative Network Models to Advance Integrative Assessments of Marine Ecosystems. *Coast.*
 *Manage.* **44**, 486-503, doi:10.1080/08920753.2016.1208881 (2016).
- Wells, B. K. *et al.* Implementing Ecosystem-Based Management Principles in the Design of a
 Salmon Ocean Ecology Program. *Frontiers in Marine Science* **7**, doi:10.3389/fmars.2020.00342
 (2020).
- Adams, J. *et al.* A century of Chinook salmon consumption by marine mammal predators in the
 Northeast Pacific Ocean. *Ecological Informatics* **34**, 44-51,
 doi:<https://doi.org/10.1016/j.ecoinf.2016.04.010> (2016).
- Thorne, K. *et al.* U.S. Pacific coastal wetland resilience and vulnerability to sea-level rise. *Science*
 *Advances* **4**, eaao3270, doi:10.1126/sciadv.aao3270 (2018).
- Kaplan, I. C. *et al.* Impacts of depleting forage species in the California Current. *Environmental*
 *Conservation* **40**, 380-393, doi:10.1017/S0376892913000052 (2013).

- Collie, J. S. *et al.* Ecosystem models for fisheries management: finding the sweet spot. *Fish and*
*Fisheries* **17**, 101-125, doi:10.1111/faf.12093 (2016).
- Skalski, J. R. *et al.* Status after 5 Years of Survival Compliance Testing in the Federal Columbia
River Power System (FCRPS). *North American Journal of Fisheries Management* **36**, 720-730,
doi:10.1080/02755947.2016.1165775 (2016).
- Welch, D. W. *et al.* Survival of Migrating Salmon Smolts in Large Rivers With and Without Dams.
*PLoS biology* **6**, 2101-2108, doi:10.1371/journal.pbio.0060265 (2008).
- Environmental Protection Agency, U. S. A. R. Total Maximum Daily Load (TMDL) for Temperature
in the Columbia and Lower Snake Rivers, May 18, 2020 TMDL for Public Comment. (2020).
- Gosselin, J. L. & Anderson, J. J. Combining Migration History, River Conditions, and Fish
Condition to Examine Cross-Life-Stage Effects on Marine Survival in Chinook Salmon.
*Transactions of the American Fisheries Society* **146**, 408-421,
doi:10.1080/00028487.2017.1281166 (2017).
- Scheuerell, M. D., Zabel, R. W. & Sandford, B. P. Relating juvenile migration timing and survival
to adulthood in two species of threatened Pacific salmon (*Oncorhynchus* spp.). *Journal of*
*Applied Ecology* **46**, 983–990 (2009).
- Zabel, R. W. & Williams, J. G. Selective mortality in chinook salmon: What is the role of human
disturbance? *Ecological Applications* **12**, 173-183 (2002).
- Bond, M. H., Nodine, T. G., Beechie, T. J. & Zabel, R. W. Estimating the benefits of widespread
floodplain reconnection for Columbia River Chinook salmon. *Canadian Journal of Fisheries and*
*Aquatic Sciences* **76**, 1212-1226, doi:10.1139/cjfas-2018-0108 (2019).
- Herbold, B. *et al.* Managing for salmon resilience in California’s variable and changing climate.
*San Francisco Estuary and Watershed Sciences* **16**, doi:Retrieved from
<https://escholarship.org/uc/item/8rb3z3nj> (2018).
- Chittaro, P. *et al.* Variability in the performance of juvenile Chinook salmon is explained
primarily by when and where they resided in estuarine habitats. *Ecology of Freshwater Fish* **27**,
857-873, doi:10.1111/eff.12398 (2018).
- Beechie, T. *et al.* Restoring salmon habitat for a changing climate. *River Research and*
*Applications* **29**, 939-960, doi:10.1002/rra.2590 (2013).
- Idaho Department of Fish and Game, Oregon Department of Fish and Wildlife & Washington
Department of Fish and Wildlife. *Snake River ESU Spring Summer Chinook Natural Origin*
*Spawner Abundance Dataset (1949-2017)*, 2018).
- Nez Perce Tribe East Fork South Fork Salmon River summer Chinook and Secesh River summer
Chinook, Natural Origin Spawner Abundance Dataset (1957-2017). (Protocol and methods
available at <https://www.cbfish.org/Document.mvc/Viewer/P165414>. Personal communication
with Mari Williams, NOAAF NWFSC/OAI 2019, 2019).
- StreamNet. Fish Data for the Northwest. (<http://www.streamnet.org/>, accessed Nov 2018,
2018).
- Faulkner, J. R., Widener, D. L., Smith, S. G., Marsh, T. M. & Zabel, R. W. Survival estimates for the
passage of spring migrating juvenile salmonids through Snake and Columbia River dams and
reservoirs, 2017. (Draft report of the National Marine Fisheries Service to the Bonneville Power
Administration. Portland, Oregon. Available at
https://www.nwfsc.noaa.gov/contact/display_staffprofilepubs.cfm?staffid=1524, 2018).
- Lamb, J. J. *et al.* Monitoring the migrations of wild Snake River spring/summer Chinook salmon
juveniles: survival and timing, 2017. (Report of the National Marine Fisheries Service to the
Bonneville Power Administration. Portland, Oregon. Available at
https://www.nwfsc.noaa.gov/contact/display_staffprofilepubs.cfm?staffid=550, 2018).

Crozier, L., Dorfmeier, E., Marsh, T., Sandford, B. & Widener, D. Refining our understanding of
early and late migration of adult Upper Columbia spring and Snake River spring/summer
Chinook salmon: passage timing, travel time, fallback and survival. (Report of research by Fish
Ecology Division, Northwest Fisheries Science Center. Available at
https://www.nwfsc.noaa.gov/contact/display_staffprofilepubs.cfm?staffid=1471, 2016).

DART. Columbia River Data Access in Real Time.
<http://www.cbr.washington.edu/dart/dart.html>, accessed June 2019 (2019).

NOAA Fisheries. Salmon Population Summary. [https://catalog.data.gov/dataset/sps-abundance-](https://catalog.data.gov/dataset/sps-abundance-salmon-population-summary-database)
[salmon-population-summary-database](https://catalog.data.gov/dataset/sps-abundance-salmon-population-summary-database), accessed July 2018 (2019).

Kareiva, P., Marvier, M. & McClure, M. Recovery and management options for spring/summer
Chinook salmon in the Columbia River basin. *Science* **290**, 977-979 (2000).

Hartig, F., Calabrese, J. M., Reineking, B., Wiegand, T. & Huth, A. Statistical inference for
stochastic simulation models – theory and application. **14**, 816-827, doi:10.1111/j.1461-
0248.2011.01640.x (2011).

Csillery, K., Blum, M. G. B., Gaggiotti, O. E. & Francois, O. Approximate Bayesian Computation
(ABC) in practice. *Trends in Ecology and Evolution* **25**, 410-418, doi:10.1016/j.tree.2010.04.001
(2010).

Gompertz, B. On the nature of the function expressive of the law of human mortality, and on a
new mode of determining the value of life contingencies. *Philosophical Transactions of the Royal*
*Society of London B: Biological Sciences* **182**, 513–585,
doi:<https://doi.org/10.1098/rstl.1825.0026> (1825).

Gelman, A., Carlin, J. B. & Rubin, D. B. *Bayesian data analysis*. (Chapman & Hall, 2004).

Vehtari, A., Gelman, A. & Gabry, J. Practical Bayesian model evaluation using leave-one-out
cross-validation and WAIC. *Statistics and Computing* **27**, 1413-1432, doi:doi: 10.1007/s11222-
016-9696-4 (2017).

Zabel, R. W., Burke, B. J., Moser, M. L. & Peery, C. Relating dam passage of adult salmon to
varying river conditions using time-to-event analysis. *American Fisheries Society Symposium* **61**,
153-163 (2008).

Crozier, L. G. *et al.* How a changing climate affects survival of Snake River Chinook and Sockeye
salmon during their upstream migration: recent extremes and likely futures *Global Change*
*Biology* (in review).

U.S. Army Corps of Engineers (ACOE), Northwestern Division Bureau of Reclamation &
Administration, P. N. R. B. P. Columbia River System Operations Draft Environmental Impact
Statement, February 2020. *DOE/EIS-0529* (2020).

Kristensen, K., Nielsen, A., Berg, C. W., Skaug, H. & Bell, B. M. TMB: Automatic Differentiation
and Laplace Approximation. *Journal of Statistical Software* **70**, 1-21, doi:10.18637/jss.v070.i05
(2016).

NOAA, National Oceanic and Atmospheric Administration,. NOAA Earth System Research
Laboratory, Climate Change web portal, CMIP5 maps. Available from
esrl.noaa.gov/psd/ipcc/ocn/ccwp.html (November 2018). (2018).

Brady, R. X., Alexander, M. A., Lovenduski, N. S. & Rykaczewski, R. R. Emergent anthropogenic
trends in California Current upwelling. *Geophysical Research Letters* **44**, 5044-5052,
doi:10.1002/2017gl072945 (2017).

Yearsley, J. R. A semi-Lagrangian water temperature model for advection-dominated river
systems. *Water Resources Research* **45**, W12405, doi:10.1029/2008wr007629 (2009).

Brekke, L., Kuepper, B. & Vaddey, S. Climate and Hydrology Datasets for Use in the RMJOC
Agencies' Longer-Term Planning Studies: Part 1 - Future Climate and Hydrology Datasets,
available at <https://www.usbr.gov/pn/climate/planning/reports/index.html>. (2010).

Reviewer #1 (conservation ecology and climate change):

This manuscript describes an effort to use a state-of-the-art life-cycle model to quantify the risks of extinction for 8 populations of Chinook salmon in the Pacific Northwest of the USA. The model relies heavily on an extraordinary dataset of individually tagged fish to estimate various functional relationships that describe the processes that affect survival throughout the life-cycle of this species. The model is used to simulate the potential effects of climate change on the viability of these 8 populations. The primary conclusion reached by these efforts is that climate warming is likely to doom this group of populations and that the primary survival bottleneck is during the fish's first year of life in the sea. While the model development is commendable and the data used are impressive, the conclusions that are drawn are not particularly new and have been appreciated for some time. Thus, it is not really clear what the main contribution from this paper actually is.

We revised the introduction to clarify three specific contributions of this model on LN50-126 as follows:

Retrospective analyses also show strong relationships between climate indices and salmon performance (e.g., 8). Looking toward the future, indirect and qualitative assessments point to anthropogenic climate change as an additional overriding threat for salmon in the North Atlantic and California Current (e.g., 9, 10-12). How to mitigate for this threat is therefore a primary concern among conservation organizations and management agencies.

... This is a novel approach to downscaling climate projections in multiple environments.

Thus, there are three reasons why existing approaches for modeling the biological impacts of climate change are inadequate for evaluating potential management actions for salmon. Similar limitations apply to other species that are migratory or have complex life histories. First, proposed management actions are usually focused on conditions in freshwater, so accounting for “carry over” effects from freshwater to marine life stages is essential for their evaluation. Carryover effects occur when an individual's previous history affects its performance in a subsequent life stage (24). For example, the timing of migration from freshwater to the ocean and back again is a key determinant of salmon survival in every life stage, and one of the most sensitive traits in relation to climate (25-28). Timing is also a key element in multiple management actions, especially those involving the hydrosystem (29) and fisheries (30). Quantification of carryover effects that will be affected by climate change is therefore essential for evaluating the net benefits of proposed actions to protect endangered species.

Second, current models of survival in the salmon marine stage rely on climate indices that cannot be linked directly to global climate model (GCM) projections, so it is impossible to conduct formal analyses of how alternative carbon emission scenarios or other anthropogenic actions to mitigate climate change might affect the timing of declines in marine survival. Nor is it possible to quantify uncertainty in modelled projections across GCMs and thus take full advantage of the Coupled Model Intercomparison Project, which represents the major advances of global climate modeling in recent decades (31, 32).

Third, approaches that are currently available for accounting for climate impacts on freshwater and marine life stages use independent downscaling methods for the two environments. Terrestrial downscaling methods usually employ statistical or dynamical downscaling of temperature and

precipitation that feed into hydrological models. Statistical downscaling is an efficient way to explore many alternative climate projections and characterize model uncertainty at many steps in the modeling chain (33). A common approach to marine downscaling, on the other hand, is to integrate GCM output into Regional Ocean Models, which in practice are only available for very few GCM projections (32). As a result, these methods often rely on projections from different GCMs and are not consistent in characterizing potential model biases, and thus uncertainty in climate projections. Moreover, they are not temporally linked, which prevents complete accounting for carryover effects from one life stage to the next.

We address each of these difficulties by developing a novel modeling approach with a flexible and explicit mechanism for accounting for the correlation structure among all climate drivers. We also use a multi-model approach to indirectly account for a change in the relationship between climate drivers and ecological responses. Finally, we allowed the timing of the initiation of juvenile migration to vary with environmental conditions, which subsequently affected both smolt migration survival and the probability that fish would be transported in a barge through the hydrosystem (a mitigation action that has fixed start and stop dates). Three factors -- timing, hydrosystem operations and transportation -- subsequently affected arrival timing at the Columbia River estuary, ocean survival, and upstream migration timing and survival. Although the details of the migration models are unique to this system, the need to account for carryover effects and the correlation structure of climate drivers in multiple environments is shared by many migratory species.

My concerns for the current manuscript are given below.

1) The manuscript attempts to grasp at jargon to increase its general appeal, but these attempts are often a bit off-target. For example, there has been a lot of interest in ‘non-stationarity’ in ecology and in fisheries science recently and this manuscript attempts to link in to these interests. However, while this term does refer to a changing variable (in the strictest statistical sense), use of this term in ecology typically is referring to changes in a relationship between variables (i.e. the relationship or correlation among variables is non-stationary). Thus, why call climate change ‘stationary’ or ‘non-stationary’? We have known for decades that climate is non-stationary; what is interesting is that the relationships between climate conditions and ecological processes are non-stationary. Thus, use of non-stationary/stationary in this manuscript is somewhat distracting. While this is just a semantic issue, I don’t think the paper benefits from the current use of these terms.

We acknowledge that we unfortunately used the term ‘stationarity’ in both of the meanings mentioned (“strictest statistical sense” and “typical ecology” sense). We replaced the former use of the word with the term “detrended” throughout. We also used more specific language for the latter use.

2) Similarly, the manuscript refers to ‘aggressive’ warming scenarios. This also seems

misplaced. Typically we refer to aggressive scenarios of curtailing carbon emissions (i.e. it's not the climate that is aggressive, it's the policy actions to reduce emissions that are aggressive).

LN 300 We have changed “more aggressive warming scenarios” to “the upper quartile of GCM projections, in which warming occurred at a faster rate”

3) The key results are expressed as the time to quasi-extinction for each of these populations. In general, I do not think this is the best way to present the key results. When a modeled population goes extinct in these simulations is based on arbitrary population thresholds that probably don't apply well when populations are reduced to very low numbers where stochastic processes are more likely to ultimately determine whether they go extinct or not. Thus, the primary results of these simulations would be more useful if the population growth rate was the response variable used to explore the consequences of different climate scenarios (i.e. the posterior distribution of lambda). The result could then be focused on how much of the posterior distribution was <1 , thereby leading to population decline, etc. Time to extinction is too arbitrary and too 'loaded' a variable that is easily misinterpreted that it shouldn't be used in these types of analyses.

We respectfully disagree with the reviewer's claim that lambda is an appropriate description of the dynamics in populations that are density dependent, such as these. Lambda changes systematically as density dependence is reduced, which results in changes in lambda being an underestimate and hence misleading representation of deterministic population declines. It also can increase when populations stabilize, but if the new level of abundance is extremely low, then this too is misleading.

So although it is the case that lambda was more negative, and a larger proportion of the posterior distribution was below 1 in the climate change scenarios, we feel this is a trivial point that is much better made with the figure showing changes in abundance.

Extinction is a process that is more closely related to the number of fish in the stream than the rate of a prolonged decline. It is exactly because stochastic processes are so important for small populations (unlike lambda), that we use this quasi-extinction threshold. Although the actual threshold is indeed arbitrary, it is based on evolutionary theory, and in particular the relationship between effective population size and raw abundance in salmon. It was developed specifically in response to the concerns mentioned by the reviewer.

We are not concerned that it is “loaded” because it is one metric that is used in formal management decisions (NMFS 2020). It is related to recovery targets and a large body of work on population viability.

To address the reviewers concerns, we re-ordered the results section to emphasize changes in abundance before introducing the concept of the QET, and modified the presentation of QET as follows:

LN 51 “With a warming climate, deterministic declines inevitably lead to extinction unless some ecological, evolutionary, or climatic rescue effect occurs (38). Climate trajectories did level off in the RCP 4.5 scenarios in the second half of the 21st century, which reduced the rate of population declines in

that scenario. However, for the most part populations had already reached very low abundances at that point.

For practical purposes in salmon management, populations that have fewer than 50 spawners on average for 4 years in a row are considered to be at extremely high risk of extinction from chance fluctuations in abundance, compensatory processes, and long-term consequences from loss of genetic variability (39, 40). The evolutionary theory behind this threshold applies to isolated populations, and these populations are not truly isolated. So some small populations may be sustainable within a larger salmon metapopulation. Nonetheless, when the majority of populations within the ESU pass this threshold, the ESU itself is at high risk. This ESU is already threatened with extinction because of historical declines (41), so although this exact threshold is somewhat arbitrary, it is a useful metric for demonstrating the severity of the declines across all of our simulations. We assessed the first year (if any) in which a population in a given simulation fell below a quasi-extinction threshold of adult abundance (QET50). The QET50 is passed when the running mean of spawners, measured at the spawning stream, drops below 50 individuals over any 4-year period (42).”

4) Line 24, ‘species’ should be ‘populations’

LN 23 We replaced “multiple species of Chinook salmon” to “multiple species of salmon”

5) ~ Line 121, it is not clear what the goals of the paper are. Is it to construct this model? The paper would benefit from some clear statements about what the primary objectives of this exercise are.

See the above changes to the introduction

6) It is surprising that the model suggests that the populations are essentially at no risk of going extinct given the ‘stationary’ climate scenario. This does not seem realistic given the current status of these populations.

Text added:

LN 261 Some small populations may be sustainable within a larger salmon metapopulation. Nonetheless, when the majority of populations within the ESU pass this threshold [QET50], the ESU itself is at high risk. This ESU is already threatened with extinction because of historical declines (41), so although this exact threshold is somewhat arbitrary, it is a useful metric for demonstrating the severity of the declines across all of our simulations.

7) I think the ‘Caveats’ section should be part of the Discussion of the paper.

That section was moved to the Discussion, LN 426.

Discussion

8) The Discussion highlights the weakness of the paper. Here the text basically runs through what is known about different processes or conditions that affect salmon survival. Nothing is particularly novel in this summary, nor does it highlight the main contributions of this specific modeling effort. Thus, it reinforces the lack of clarity about what this specific research activity contributes to our general understanding of Chinook salmon ecology and conservation.

We modified the discussion to note the specific benefit of quantifying climate impacts in this study:

LN 399 Our analysis showed relative resilience in freshwater stages, with the dominant driver towards extinction being rising SST, which tracked a ~90% decline in survival in the marine life stage. This occurred despite an advance in smolt migration timing and other changes in hydrosystem management. The modeled carryover effects of changes in timing are likely to be adaptive, but inadequate as compensation for large declines in marine survival.

And

LN 454 The results of our model 3, in which marine survival was driven by upwelling indicated that improved productivity in a warmer ocean could benefit salmon. Nonetheless, the benefit in that case was relatively small compared with overall negative effects.

And

LN 620 Our modeling approach, which accounts for carryover effects across the life cycle, allows systematic exploration of alternative correlation structures among climate drivers and between climate drivers and ecological responses, and a thorough accounting for uncertainty in climate projections lays a path forward for evaluating benefits of proposed actions to protect our critical resources.

Reviewer #3 (Remarks to the Author):

In this paper the authors apply a stage-based life history model to 8 populations of Chinook salmon in the Columbia River/Snake River basins and show strong associations between warming (particularly warming SST) during the marine stage of the life history and probability of population extinction/extirpation. I accepted the review with considerable excitement of seeing something truly new, insightful, or transformative. Unfortunately I was underwhelmed, not by the statistical rigour, but rather with the interpretation of the results. The claim is essentially that warming is bad and that smolts need to survive the ocean better for populations to avoid extinction. The authors all but said, and perhaps should have, that this analysis provides strong evidence that these populations are doomed and that restoration/conservation is a fool's errand (that would indeed have been provocative at least).

What I was hoping to see more of was a more holistic linkage between different stages of the life history and a quantitative appreciation that what happens in freshwater may lead individuals down trajectories that result in the ocean life history being the proximate stage of mortality.

We added more specific details about the changes in timing.

LN 363 Our model predicted that smolts would shift their migration timing about 4.5 days earlier arrival at Lower Granite Dam, which does reduce temperature exposure. Nonetheless, temperature effects on the juvenile migration still grew over time, and reduced populations by about -18% on average from the 2020s to the 2060s.

Climate impacts were most dramatic in the marine stage, where survival was reduced by -83 to -90% (Fig. 6). This occurred despite the fact that smolts arrived at Bonneville Dam, initiating the marine stage, about 6.5 days earlier, which generally improves marine survival (26).

Adult Chinook were predicted to shift their migration ~4 days earlier in response to warmer mainstem conditions with lower flows (43). But again, this was not enough to prevent mortality from increased heat exposure. During the adult migration, populations that returned to their spawning areas in summer (Secesh River and Valley Creek) were more affected by temperature than spring-run populations, with net declines of up to -17% by the 2060s. Still, the declines we found in the mainstem were relatively small because of the early adult run timing of Snake River spring/summer Chinook compared with other run types or species that migrate during peak temperatures.

And in the discussion:

LN 404 The modeled carryover effects of changes in timing are likely to be adaptive, but inadequate as compensation for large declines in marine survival, regardless of entry timing

I was hoping and expecting to see discussion that warming SSTs may be detrimental to southern Chinook populations, but Alaska populations may fare better during warming temperatures (and indeed the authors did not cite the obvious paper to suggest so).

Text added:

LN 419 Some Alaskan populations may fare better in response to ocean warming, but negative responses to warming in freshwater have been observed in other Alaskan populations, highlighting the benefits of an examination of the full life cycle (Cunningham et al. 2018; Jones et al. 2020; Ohlberger et al. 2016).

And

LN 561 Reducing hatchery production could have positive benefits, especially for wild Alaskan salmon which might benefit from warmer conditions in some localities (Cunningham et al. 2018; Jones et al. 2020).

So in the end I am left wanting the authors to make a better case for novelty and insights that can be gleaned by this very complex modelling exercise that goes beyond what is already firmly established.

Text added:

LN 422 Quantitative comparisons of combined climatic and anthropogenic influences and more exploration of changing relationships among drivers are needed to unravel the multiple pressures these populations face

AND

LN 514 Additional changes to the juvenile transportation schedule or faster transit through the mainstem can still be explored, but additional improvements to survival through other mechanisms are needed.

AND

LN 620 Our modeling approach, which accounts for carryover effects across the life cycle, allows systematic exploration of alternative correlation structures among climate drivers and between climate drivers and ecological responses, and a thorough accounting for uncertainty in climate projections lays a path forward for evaluating benefits of proposed actions to protect our critical resources.

Although the reference section is extensive, I do suggest the authors incorporate information from populations beyond their focal range to broaden the discussion of SSTs and to also contrast their work to other very similar approaches.

Cunningham, Curry J., Peter AH Westley, and Milo D. Adkison. "Signals of large scale climate drivers, hatchery enhancement, and marine factors in Yukon River Chinook salmon survival revealed with a Bayesian life history model." *Global change biology* 24.9 (2018): 4399-4416.

And a new paper just out led by Leslie Jones (open access)

<https://onlinelibrary.wiley.com/doi/full/10.1111/gcb.15155>

References added

Reviewer #2 (Remarks to the Author):

I found this paper to be exceptionally well written and the topic to be of broad international interest as a study of the impact of climate on a commercially valued species that is now at risk. This work explores the population dynamic of 8 populations of Chinook salmon in the Snake River watershed. This paper uses a life cycle model and two climate change scenarios to model the potential population trajectories of these populations. It effectively tells the risks of changing climate, particularly at sea, to all these populations but especially the smaller ones. It is sobering in its assessment of the viability of these populations- not a happy story. However, it effectively looks at the fact that these fish persist in highly altered systems and actions in freshwater might mitigate or at least slow population declines and localized extinction. These findings are novel for the populations of interest and findings (and solutions) will they be of interest to not only the fisheries

biology community but ecologists and also to the public. This work is a tangible story of the impact of change upon a species that captures the imagination of a broad spectrum of society. The species also has rich cultural importance to first nations in this watershed. These conclusions are supported by appropriate modeling and add to not only Pacific salmon literature but will be of great interest to Atlantic salmon conservationists. The sea-run nature of this species also will inform understanding of challenges in freshwater and marine systems. I found this paper to be extremely convincing and the conclusions are strengthened by a rich use of citations and the addition of allied related manuscripts on the stream temperature model and wild/hatchery marine survival. This work is cutting edge and current. I firmly believe that this paper will influence thinking in the field due to its clarity of message and exceptional graphical presentation of results. I am generally familiar with salmon life cycle models and the overall use was both appropriate and added novel methods. I noted in one part of the paper where I am less familiar with one of the statistical analyses used. I would defer to you and others there. The open nature of the models, data and coding would allow not only the researchers to reproduce the work but add on to a toolkit of models that could be used for other species and habitats.

I have included comments in the margin of the manuscript as well as made some direct in-text suggestions to improve clarity. This was a pleasure to read.

John F. Kocik

Thank you!

Specific comments in text:

We edited the text following nearly all of the minor editorial recommendations made in tracked changes, which are not repeated here. I pulled out the comments and discuss how we addressed them.

Ln 54 ¹⁹Atlantic Multidecadal Oscillation (AMO))

Reference added to AMO and Mills et al 2013:

Although marine climate indices have been tightly linked to survival (e.g., Pacific Decadal Oscillation (PDO) (19) and the Atlantic Multidecadal Oscillation (AMO)(20)),...

LN 87 We applied the model to eight populations within the Snake River spring/summer Chinook salmon Evolutionarily Significant Unit (ESU)

The target audience for the introduction consists of people who are not familiar with these specific locations, so the names of populations would not be informative for the general audience. The population names are included in Fig. 2, which is in the introduction and should therefore be quickly accessible when the paper is typeset.

LN 112 interactions between atmospheric forcing, wind strength, upwelling, and mixing of ocean layers, all of which affect productivity throughout the California Current Ecosystem²⁸

Added text: ecosystem productivity

LN 121 Although small populations drop below this threshold periodically even in a stationary climate, larger populations do not.

All of these populations are certainly much smaller than they were historically, but we suspect their relative ordering was similar to today.

We modified the text as follows:

Some small populations may be sustainable within a larger salmon metapopulation. Nonetheless, when the majority of populations within the ESU pass this threshold, the ESU itself is at high risk. This ESU is already threatened with extinction because of historical declines (41), so although this exact threshold is somewhat arbitrary, it is a useful metric for demonstrating the severity of the declines across all of our simulations.

LN 125 ... even the largest populations (Bear Valley Creek and Secesh River) fell below QET50\

We added a figure showing population dynamics over the past 20 years, including my model simulation of those dynamics, and stated the geometric mean population size at the beginning of the Methods section as follows:

Geometric mean population sizes from 2005-2014 in the populations shown in Fig. 2 ranged from 38 to 474 (39)

We feel that the names and locations of the populations in the figures is sufficient for a general audience. Detailed spawning maps do not seem appropriate for this manuscript.

LN 142 Are data available to have 2000 or 2010 starts with actual abundance metrics for past 10 to 20 years? Would enhance reader understanding of initial states. (Legend for Fig. 4)

We added a figure in the Methods section (fig. 8) showing historical population dynamics, and refer to this section in the first paragraph of the Results section:

For understanding of recent population dynamics and historical population modeling, see Methods.

LN 168 Fig. 5.

We added this information to the introduction:

We simulated population time series for these exclusively wild populations, whose recent mean spawner abundance ranged from 45 to near 600.

LN 218 More generally, the high-elevation, mostly-wilderness habitat of these populations is unusual for salmon in the region, and partially explains the relatively small effects of climate change on their freshwater life stages.

Citation added: Paulsen and Fisher 2001

LN 227 Therefore, closely monitoring ocean survival and directing research into these populations potential response to novel conditions is clearly needed.

This sentence was removed during general editing

LN 253 closure of multiple fisheries in 2020.

This sentence was removed during general editing

LN 272 Small populations had minimal ability to buffer against declining marine survival rates, and thus were at the most immediate risk (Fig. 1).

Changed to:

Small populations had minimal demographic buffers against declining marine survival rates, and thus quickly dropped below the quasi-extinction threshold in nearly all simulations (Fig. 5).

LN 295 Do these populations use this type of [estuary] habitat? Try not to get too broad and make sure linking to the populations in this paper.

Text added:

Although the populations studied here generally spend relatively little time in the Columbia estuary, smaller, earlier-migrating individuals do utilize these habitats (Weitkamp et al. 2012). As fish outmigrate earlier in the future, they might rely more on estuary habitat. Furthermore, other salmon populations depend heavily on estuary habitat for essential growth and their success has been linked to estuary restoration actions (Diefenderfer et al. 2016).

LN 312 also enhance/restore freshwater salmon production.

Changed as suggested

LN 320 sport fish (smallmouth bass, *Micropterus dolomieu* and brook trout *Salvelinus fontinalis*) to creation of reservoir habitat more favorable for invasive fish (e.g., American shad, *Alosa sapidissima*).

We removed the Latin names here for consistency

LN 326 habitat

The word “prime” was changed to “all critical”

LN 333 However, the urgency is greater than ever to identify successful solutions at a large scale and implement known methods for improving survival. Management actions that open new habitat, improve productivity within existing habitat, or reduce mortality through direct or indirect effects in the ocean are desperately needed.

Text added: However, there are hard choices where human demands on land and water have come at the cost of wildlife. The urgency is greater than ever to identify successful solutions at a large scale and implement known methods for improving survival. Management actions that open new habitat, improve productivity within existing habitat, or reduce mortality through direct or indirect effects in the ocean are desperately needed. We can find new ways to improve salmon habitats while maintaining other benefits for people, like reconnecting floodplains for recharging aquifers, and protection from flooding, storm surge and erosion with floodplains and natural marshes.

Methods section:

LN 339 abundance estimates .

Text added to introduction:

LN 154 We simulated population time series for these exclusively wild populations, whose recent 15-y geometric mean spawner abundance ranged from 45 to near 600 (Ford et al. 2016; NMFS 2020) and shown in new Fig. 8.

wild from known population sources.

Text added:

LN 632 No hatchery fish are released in any of the streams supporting the populations in this analysis.

“effective spawners”,

Thank you

of producing a spawner distribution that was not statistically different from the observation dataset (p -value >0.05 from the Kolmogorov-Smirnov test).

Thank you

The COMPASS model also tracks the proportion of fish that were loaded into barges to bypass migration through the hydropower system.

Text added to Supplement S2:

LN 58 The COMPASS model predicts the proportion of fish passing a dam that will enter the bypass system as a function of percent spill, flow, and potentially also day of year or water temperature (depending on the dam).

(500,000 iterations per population).

Please note that the formal convergence tests were conducted on the Bayesian mcmc chain output, as described in the supplement S1. We added more information to that section:

LN 105 We assessed convergence of the chains using the Gelman and Rubin’s convergence diagnostic (gelman.diag function in the coda package). The multivariate potential scale reduction factor was <1.0125 for all initial models (6 models, in which covariates included summer temperature and one of spring, summer or fall flow, and covariates were incorporated into either the productivity or the capacity terms). We also examined Heidelberger and Welch’s convergence diagnostics. To ensure all chains were long enough, we re-ran all models with a single chain that was 15 million iterations. All of the parameters in all models passed this diagnostic, except for the two models that included both summer temperature and summer flow. They still had one parameter each that failed the Heidelberger

and Welch convergence diagnostic (at $\text{eps}=0.1$ and $\text{pvalue}=0.05$). Although visual examination of the chains and density distributions looked satisfactory, we did not use these models in further analysis.

But for the calibration step, we assessed goodness of fit using the Kolmogorov-Smirnov test.

Text added:

LN 770 We selected this number so that we ended up with at least 1000 sets of parameter values for each population that produced spawner distributions similar to that observed.

REVIEWERS' COMMENTS:

Reviewer #2 (Remarks to the Author):

I was quite impressed by the first version of this manuscript. I find this manuscript continued to build upon the existing salmon literature to provide both an expanded and nuanced view of climate impacts on salmon. Results are appropriate and important to share with a broader community. I understand and appreciate some of the concerns relative to focus of the paper raised by other reviews. I believe that the revisions strengthened and further clarified. I would also point out that this paper is of broad geographic reach as some of the findings are applicable to Atlantic salmon and probably other searun fish such as shad and eels. I had no "in-line" comments in this second review and support all the changes made by the authors. All my minor suggestions were either adequately addressed or rebuttal explanations were clear and appropriate. It was a pleasure to reread.

John F. Kocik

Reviewer #3 (Remarks to the Author):

The authors have addressed my most substantive concerns and clarified the novelty of their work (forecasting rather than explanation/hindcasting). This is an important, albeit fairly depressing look into the future, that I predict will shape conversations about conservation/restoration actions given the predictions of warming SSTs.